# The Ross Sea Dipole - Temperature, Snow Accumulation and Sea Ice Variability in the Ross Sea Region, Antarctica, over the Past 2,700 Years

**RICE Community**

(Nancy A.N. Bertler[1,2], Howard Conway[3], Dorthe Dahl-Jensen[4], Daniel B. Emanuelsson[1,2], Mai Winstrup[4], Paul T. Vallelonga[4], James E. Lee[5], Ed J. Brook[5], Jeffrey P. Severinghaus[6], Taylor J. Fudge[3], Elizabeth D. Keller[2], W. Troy Baisden[2,7], Richard C.A. Hindmarsh[8], Peter D. Neff[1,2,9], Thomas Blunier[4], Ross Edwards[10,30], Paul A. Mayewski[11], Sepp Kipfstuhl[12], Christo Buizert[5], Silvia Canessa[2], Ruzica Dadic[1], Helle A. Kjær[4], Andrei Kurbatov[11], Dongqi Zhang[13,14], Edwin D. Waddington[3], Giovanni Baccolo[15], Thomas Beers[11], Hannah J. Brightley[1,2], Lionel Carter[1], David Clemens-Sewall[16], Viorela G. Ciobanu[4], Barbara Delmonte[15], Lukas Eling[1,2], Aja Ellis[17], Shruthi Ganesh[18], Nicholas R. Golledge[1,2], Skylar Haines[11], Michael Handley[11], Robert L. Hawley[16], Chad M. Hogan[19], Katelyn M. Johnson[1,2], Elena Korotkikh[11], Daniel P. Lowry[1], Darcy Mandeno[1], Robert M. McKay[1], James A. Menking[5], Timothy R. Naish[1], Caroline Noerling[12], Agathe Ollive[20], Anaïs Orsi[21], Bernadette C. Proemse[19], Alexander R. Pyne[1], Rebecca L. Pyne[2], James Renwick[1], Reed P. Scherer[22], Stefanie Semper[23], M. Simonsen[4], Sharon B. Sneed[11], Eric J., Steig[3], Andrea Tuohy[1,2,24], Abhijith Ulayottil Venugopal[1,2], Fernando Valero-Delgado[12], Janani Venkatesh[18], Feitang Wang[25], Shimeng Wang[14], Dominic A. Winski[16], V. Holly L. Winton[26], Arran Whiteford[27], Cunde Xiao[28], Jiao Yang[14], Xin Zhang[29])

[1] Antarctic Research Centre, Victoria University of Wellington, Wellington, 6012, New Zealand
[2] GNS Science, Lower Hutt, 5010, New Zealand
[3] Department of Earth and Space Sciences, University of Washington, Seattle, WA 98195, USA
[4] Centre for Ice and Climate, Niels Bohr Institute, University of Copenhagen, Juliane Maries Vej 30, 2100 Copenhagen, Denmark
[5] College of Earth, Ocean, and Atmospheric Sciences, Oregon State University, Corvallis, OR 97330, USA
[6] Scripps Institution of Oceanography, UC San Diego, La Jolla CA 92093, USA
[7] Now at Faculty of Science and Engineering, University of Waikato, Hamilton, New Zealand
[8] British Antarctic Survey, High Cross, Madingley Road, Cambridge, CB3 0ET, United Kingdom
[9] Now at University of Rochester, Department of Earth & Environmental Sciences, Rochester, NY, USA 14627
[10] Physics and Astronomy, Curtin University, Perth, Western Australia, Australia
[11] Climate Change Institute, University of Maine, Orono, ME 04469-5790, USA
[12] Alfred Wegner Institute, Bremen, Germany
[13] Chinese Academy of Meteorological Sciences, Beijing, China
[14] State Key Laboratory of Cryospheric Science, Northwest Institute of Eco-Environment and Resources, Chinese Academy of Sciences, Lanzhou, Gansu, China
[15] DISAT, Department of Earth and Environmental Sciences, University Milano-Bicocca, Piazza della Scienza 1, 20126 Milano (Italy)
[16] Department of Earth Sciences, Dartmouth College, 6105 Fairchild Hall, Hanover, NH 03755, USA
[17] Physics and Astronomy, Curtin University, Perth, Western Australia, Australia; Now at Center for Atmospheric Particle Studies, Carnegie Mellon University, Pittsburgh, PA 15213 USA.
[18] Department of Chemical Engineering, SRM University, Kattankulathur - 603203, Kancheepuram Dt., Tamil Nadu, India
[19] University of Tasmania, School of Biological Sciences, Hobart, TAS, 7001 Australia
[20] Specialty of Earth Sciences and Environment, UniLasalle, 19 rue Pierre Waguet, 60000 Beauvais, France

21 Laboratoire des Sciences du Climat et de l'Environnement, LSCE/IPSL, CEA-CNRS-UVSQ, Université Paris-Saclay, F-91198 Gif-sur-Yvette, France

22 Northern Illinois University, USA

23 Geophysical Institute, University of Bergen, and Bjerknes Centre for Climate Research, 5020 Bergen, Norway

24 Now: Tonkin and Taylor, ABS Tower, 2 Hunter St., Wellington, 6011, New Zealand

25 State Key Laboratory of Cryospheric Science / Tianshan Glaciology Station, Northwest Institute of Eco-Environment and Resources, Chinese Academy of Sciences, Lanzhou, Gansu, China

26 Physics and Astronomy, Curtin University, Perth, Western Australia, Australia; now at: British Antarctic Survey, Cambridge CB3 0ET, United Kingdom

27 Department of Earth and Ocean Sciences, University of British Columbia, Canada

28 State Key Laboratory of Earth Surface Processes and Resource Ecology, Beijing Normal University, Beijing, China

29 Northwest Normal University, Lanzhou, Gansu, China

30 Now: University of Wisconsin-Madison, Department of Civil and Environmental Engineering, 660 N. Park St., Madison

*Correspondence to*: Nancy A.N. Bertler (Nancy.Bertler@vuw.ac.nz)

**Abstract.** High-resolution, well-dated climate archives provide an opportunity to investigate the dynamic interactions of climate patterns relevant for future projections. Here, we present data from a new, annually-dated ice core record from the eastern Ross Sea, named the Roosevelt Island Climate Evolution (RICE) ice core. Comparison of this record with climate reanalysis data for the 1979-2012 interval shows that RICE reliably captures temperature and snow precipitation variability of the region. Trends over the past 2,700 years in RICE are shown to be distinct from those in West Antarctica and the western Ross Sea captured by other ice cores. For most of this interval, the eastern Ross Sea was warming (or showing isotopic enrichment for other reasons), with increased snow accumulation and perhaps decreased sea ice concentration. However, West Antarctica cooled and the western Ross Sea showed no significant isotope temperature trend. This pattern here is referred to as the Ross Sea Dipole. Notably, during the Little Ice Age, West Antarctica and the western Ross Sea experienced colder than average temperatures, while the eastern Ross Sea underwent a period of warming or increased isotopic enrichment. From the 17th Century onwards, this dipole relationship changed. All three regions show current warming, with snow accumulation declining in West Antarctica and the eastern Ross Sea, but increasing in the western Ross Sea. We interpret this pattern to reflect an increase in sea ice in the eastern Ross Sea with perhaps the establishment of a modern Roosevelt Island polynya as a local moisture source for RICE.

## 1 Introduction

With carbon dioxide ($CO_2$) and global temperatures predicted to continue to rise, model simulations of the Antarctic / Southern Ocean region show for the coming decades an increase in surface warming resulting in reduced sea ice extent, weakened Antarctic Bottom Water formation, intensified zonal winds that reduce $CO_2$ uptake by the Southern Ocean, a slowing of the southern limb of the meridional overturning circulation (MOC) and associated changes in global heat transport, and rapid ice sheet grounding line retreat that contributes to global sea level rise (Kusahara and Hasumi, 2013; Spence et al., 2012; Marshall and Speer, 2012; Sen Gupta et al., 2009; Toggweiler and Russell, 2008; Russell et al., 2006; Downes et al., 2010; Anderson et al., 2009; DeConto and Pollard, 2016; Joughin and Alley, 2011; Golledge et al., 2015; DeVries et al., 2017). Observations confirm an ozone-depletion-induced strengthening and poleward contraction of zonal winds (Thompson and Solomon, 2002b; Arblaster et al., 2011), increased

upwelling of warm, modified Circumpolar Deep Water (Jacobs et al., 2011), a warmer Southern Ocean (Böning et al., 2008;Gille, 2002;Abraham et al., 2013), meltwater-driven freshening of the Ross Sea (Jacobs et al., 2002), ice shelf and mass loss, grounding line retreat (Rignot et al., 2014;Pollard et al., 2015;Paolo et al., 2015;Joughin et al., 2014), reduced formation of Antarctic Bottom Water (Rintoul, 2007) and Antarctic Intermediate Water (Wong et al., 1999), changes in sea ice (regional decreases and increases

in the Amundsen and Ross Seas, respectively) (Holland and Kwok, 2012;Sinclair et al., 2014;Stammerjohn et al., 2012), and dynamic changes of Southern Ocean $CO_2$ uptake driven by atmospheric circulation (Landschützer et al., 2015). Yet, these observational time series are short (Böning et al., 2008;Gille, 2002;Toggweiler and Russell, 2008) and inter-model variability indicates that physical processes and their consequences are not well captured or understood (Braconnot et al., 2012;Sen Gupta et al., 2009). While the skill of equilibrium simulations steadily improves, the accuracy of transient model projections for the

coming decades critically depends on an improved knowledge of climate variability, forcings, and dynamic feedbacks (Stouffer et al., 2017;Bakker et al., 2017).

Here we present data from a new, highly-resolved and accurately-dated ice-core record, spanning the past 2.7 ka, from the eastern Ross Sea region. The Roosevelt Island Climate Evolution (RICE) ice core is compared with existing records in the region to investigate the characteristics and drivers of spatial and temporal climate variability in the Ross Sea region.

## 2      Site characteristics and relevant climate drivers

In this section, a brief overview is provided of the climatological and glaciological characteristics of the study site.

### 2.1      Dynamic interaction between tropical and mid-latitudinal climate drivers and South Pacific climate variability

Environmental variability in the Pacific Sector of the Southern Ocean and Antarctica is dominated by three major atmospheric

circulation patterns: the Southern Annular Mode (SAM), the El Niño Southern Oscillation (ENSO) and the Inter-decadal Pacific Oscillation (IPO). The SAM is the leading empirical orthogonal function (EOF) of the Southern Hemisphere extratropical geopotential height fields on monthly and longer time scales, and describes the strength and position of the Southern Hemisphere westerly winds via the relative pressure over Antarctica (~65°S) and the mid-latitudes (~45°S) (Thompson and Wallace, 2000;Thompson and Solomon, 2002a). The persistent positive, summer trend of the SAM

(decreasing pressure over Antarctica) has been linked to stratospheric ozone depletion and increase in atmospheric greenhouse gas concentration (Arblaster et al., 2011;Thompson et al., 2011). The positive SAM is associated with above average warming of the Antarctic Peninsula, cooler conditions over East Antarctica due to a reduced poleward pressure gradient and thus diminished transport of heat and moisture and a reduction in katabatic flow (Marshall et al., 2013;Marshall and Thompson, 2016;Thompson and Solomon, 2002a). While the positive summer SAM trend (also weakly expressed during autumn) along

the Antarctic margin is generally associated with an equatorward heat flux, the western Ross Sea is one of two regions (the Weddell Sea being the other) to experience an anomalous poleward heat flux (Marshall and Thompson, 2016), that transports heat and moisture across the Ross Ice Shelf. No such pattern is observed for regional SAT (Marshall and Thompson, 2016), which might be masked by the influence of regional sea ice variability on local temperatures. The positive SAM has been shown to contribute at least partially to an increase in total Antarctic sea ice, while a negative SAM has been associated with

a reduced sea ice (Ferreira et al., 2015;Holland et al., 2016;Bintanja et al., 2013;Kohyama and Hartmann, 2015;Turner et al., 2017). The future behaviour of the SAM over the next decades is a topic of active research due to the competing, and seasonally biased influences of projected stratospheric ozone recovery and greenhouse gas emissions (Bracegirdle et al., 2014;Gillett and Fyfe, 2013;Thompson et al., 2011).

The Pacific South American (PSA) patterns represent atmospheric Rossby wave trains initiated by anomalously deep tropical

convection during ENSO events, in particular during austral spring, which originate in the western (PSA1) and the central

(PSA2) tropical Pacific (Mo and Higgins, 1998). PSA1 and PSA2 are defined as the 2nd and 3rd EOF respectively of monthly-mean extratropical geopotential height fields, with the negative (positive) phase resembling El Niño (La Niña)-like conditions (Mo, 2000). Changes in SAT over West Antarctica have been linked to PSA1 variability (Schneider et al., 2012;Schneider and Steig, 2008), while the warming of West Antarctica's winter temperatures has been linked to PSA2 (Ding et al., 2011). The positive polarity of PSA1 is associated with anticyclonic wind anomalies in the South Pacific centered at ~120°W, which have been linked to increased onshore flow and increased eddy activity (Marshall and Thompson, 2016). In contrast, during the positive phase of the PSA2, the anticyclonic centre shifts to ~150°W in the Ross Sea, creating a dipole across the Ross Ice Shelf, with increased transport of marine air masses along the western Ross Ice Shelf and enhanced katabatic flow along the eastern Ross Ice Shelf (Marshall and Thompson, 2016). Sea ice feedbacks to the SAM and ENSO forcing in the western Ross Sea (as well as the Bellingshausen Sea) were found to be particularly strong when a negative SAM coincided with El Niño events (increased poleward heat flux, less sea ice) or a positive SAM concurred with La Niña events (decreased poleward heat flux, more sea ice) (Stammerjohn et al., 2008). The authors found that this teleconnection is less pronounced in the eastern Ross Sea.

The Inter-decadal Pacific Oscillation (IPO), an ENSO-like climate variation on decadal time scales (Power et al., 1999), is closely related to the Pacific Decadal Oscillation (PDO) (Mantua and Hare, 2002). While the PDO is defined as the first EOF of sea surface temperature (SST) variability in the Northern Pacific, the IPO is defined by a tripole index of decadal scale SST anomalies across the Pacific (Henley et al., 2015). A warm tropical Pacific and weakened trade winds are associated with a positive IPO, while a cooler tropical Pacific and strengthened trade winds are characteristic for a negative IPO. The phasing of the IPO and PDO have been shown to influence the strength of regional and global teleconnections with ENSO (Henley et al., 2015). An in-phase IPO amplification of ENSO events has been linked to a strengthening of global dry / wet anomalies, in contrast to periods when the IPO and ENSO are out of phase, causing these anomalies to weaken or disappear entirely (Wang et al., 2014). In addition, a negative IPO leads to cooler SSTs in the Ross, Amundsen and Bellingshausen Seas, while a positive IPO is associated with warmer SSTs (Henley et al., 2015). The centre of anticyclonic circulation linked to precipitation at Roosevelt Island (Emanuelsson et al., in review) moves eastward during the negative IPO from ~120°W during the positive IPO to ~100°W (Henley et al., 2015). It has been suggested that the negative IPO, at least in part, is responsible for the hiatus of global surface warming during 1940-1975 and 2001-2009 (England et al., 2014;Kosaka and Xie, 2013;Meehl et al., 2011). The Amundsen Sea Low (ASL), a semi-permanent low pressure centre in the Ross / Amundsen Sea, is the most prominent and persistent of three low pressure centres around Antarctica, associated with the wave number 3 circulation (Raphael, 2004;Raphael et al., 2016;Turner et al., 2013). The ASL is sensitive to ENSO (especially during winter and spring) and SAM (in particular during autumn), and influences environmental conditions in the Ross, Amundsen and Bellingshausen Seas and across West Antarctica and the Antarctic Peninsula (Ding and Steig, 2013;Raphael et al., 2016;Steig et al., 2012;Marshall and Thompson, 2016;Bertler et al., 2004;Turner et al., 2013;Thomas et al., 2009). Seasonally, the ASL centre moves from ~110° W in during austral summer to ~150°W austral winter (Turner et al., 2013). A positive SAM and / or La Niña event leads to a deepening of the ASL, while a negative SAM and/or El Niño event causes a weakening (Turner et al., 2013). The IPO, through its effect upon ENSO and SAM variability, also influences the ASL and sea ice extent in the Ross and Amundsen Seas (Meehl et al., 2016). Blocking events in the Amundsen Sea (Renwick, 2005), are sensitive to the position of the ASL and are dominant drivers of marine air mass intrusions and associated precipitation and temperature anomalies at Roosevelt Island (Emanuelsson et al., in review).

## 2.2  RICE site characteristics

Roosevelt Island is an approximately 120 km-long by 60 km-wide grounded ice rise located near the north-eastern edge of the Ross Ice Shelf (Figure 1). Ice accumulates locally on the ice rise, while the floating Ross Ice Shelf flows around Roosevelt

Island. The ice surrounding Roosevelt Island originates from the West Antarctic Ice Sheet (WAIS), via the Bindschadler, MacAyeal and Echelmeyer ice streams. Bedmap2 data (Fretwell et al., 2013) suggest that the marine basins on either side of Roosevelt Island are roughly 600 m (western basin) and 750 m (eastern basin) deep and that the thickness of the Ross Ice Shelf at this location is about 500 m. At the RICE drill site (79.364 S, 161.706 W, 550 m above sea level) near the summit of Roosevelt Island, the ice is 764 m thick and grounded 214 m below sea level. Radar surveys across the Roosevelt Island ice divide show well-developed "Raymond Arches" (Raymond, 1984) of isochrones suggesting a stable ice divide (Conway et al., 1999). The vertical velocity, constrained by phase-sensitive radio echo sounder (pRES) measurements, is approximately 20 cm a$^{-1}$ at the surface relative to the velocity of 0 cm a$^{-1}$ at the bed (Kingslake et al., 2014). A small migration of the divide has occurred in the past few centuries with the topographic divide off-set by about 500 m to the southwest. It is possible that the divide migrated as a result of an imbalance in the ice flux on either side of the divide, by either changes in the snow accumulation gradient or changes in the efflux across the grounding-line due perhaps to changes in the buttressing by the Ross Ice Shelf. However, the negligible divide position migration magnitude suggests neither snow accumulation gradient nor grounding line efflux have changed very much; this implies that the buttressing has not changed significantly either.

## 3    Ice core data

During two field seasons, 2011-2012 and 2012-2013, a 764 m-long ice core to bedrock was extracted from the summit of Roosevelt Island. The drilling was conducted using the New Zealand intermediate depth ice core drill *Te Wāmua Hukapapa* ('Ice Cores That Discover the Past' in NZ Te Reo native language). The drill system is based on the Danish Hans Tausen Drill with some design modification (Mandeno et al., 2013). The upper 60 m of the borehole was cased with plastic pipe and the remainder of the drill hole filled with a mixture of Estisol-240 and Coasol to prevent closure. The part of the RICE ice core record used in this study covers the past 2.7 ka and consists of data combined from the RICE-2012/13-B firn core (0.19-12.30 m depth) and the RICE Deep ice core (12.30 m – 344 m depth). In addition, the uppermost samples from a 1.5m deep snow pit were used to extend the firn and ice core record for the month of December 2012. An overview of the core quality and processing procedures are summarised by Pyne et al. (in review). Here, we present new water stable-isotope (deuterium, δD) and snow accumulation records and compare them with existing records from West and East Antarctica (Table 1).

### 3.1    RICE age model: RICE17

The RICE17 age model for the past 2.7 ka is based on an annually-dated ice core chronology from 0-344 m which is described in detail by Winstrup et al. (2017). The cumulated age uncertainty for the past 100 years is ≤ ± 2 years, for the past 1,000 years ≤ ± 19 years and for the past 2,000 years ≤ ± 38 years, reaching a maximum uncertainty of ± 45 years at 344 m depth (2.7 ka). The RICE17 timescale is in good agreement with the WD2014 annual-layer counted timescale from the WAIS Divide ice core dating to 200 CE (280 m depth). For the deeper parts of the core, there is likely a small bias (2-3 %) towards undercounting the annual layers, resulting in a small age offset compared to the WD2014 timescale.

### 3.2    Snow accumulation reconstruction

Ice core annual layer thicknesses provide a record of past snow accumulation once the amount of vertical strain has been accounted for. At Roosevelt Island, repeat pRES measurements were performed across the divide, providing a direct measurement of the vertical velocity profile (Kingslake et al., 2014). This has a key advantage over most previous ice-core inferences of accumulation rate because vertical strain thinning through the ice sheet is measured directly, rather than needing

to use an approximation for ice-flow near ice divides (e.g. Dansgaard and Johnsen, 1969; Lliboutry, 1979). Uncertainty in the accumulation-rate reconstruction increases from zero at the surface (no strain thinning) to ± 10 % at 78 m true depth (1712 CE). Below 78 m, the uncertainty remains constant at ± 10 %. A detailed description of the accumulation-rate reconstruction is provided by Winstrup et al. (2017).

## 3.3    Water stable-isotope data

The water stable-isotope record was measured using a continuous-flow laser spectroscopy system with an off-axis integrated cavity output spectroscopy (OA-ICOS) analyser, manufactured by Los Gatos Research (LGR). The water for these measurements was derived from the inner section of the continuous flow analysis (CFA) melt head, while water from the outside section was collected for discrete samples. A detailed description and quality assessment of this system is provided by Emanuelsson et al. (2015). The combined uncertainty for deuterium (δD) at 2 cm resolution is ± 0.85‰. A detailed description of the isotope calibration, the calculation of cumulative uncertainties, and the assignment of depth is provided by Keller et al. (Keller et al., in review).

## 4    RICE data correlation with reanalysis data - modern temperature, snow accumulation, and sea ice extent trends

The ERA-Interim (ERAi) reanalysis data set of the European Centre for Medium-Range Weather Forecasts (ECMWF, Dee et al., 2011) , along with firn and ice cores, snow stakes and an automatic weather station (AWS) are used to characterise the meteorology of Roosevelt Island. ERAi data have been extracted for the RICE drill site from the nearest grid point (S 79.50°, W 162.00°, Figure 1b) for the common time period between 1979 and 2012. The year 1979, the onset of the ERAi reanalysis, occurs at 13.42 m depth in the firn. For this reason, the period 1979-2012 is predominantly captured in the RICE 12/13 B firn core. Data from the RICE AWS suggest that precipitation at RICE can be irregular, with large snow precipitation events dominating the accumulation pattern (Emanuelsson, 2016). Therefore, we limit the analysis in this study to annual averages and longer-term trends.

## 4.1    Isotope-temperature correlation

The borehole temperature measured in 2012 in two 11 m and 12 m deep drill holes suggest an average annual temperature of -23.5 °C. This stands in stark contrast to the average annual temperature derived from ERAi data of -27.4 °C at the RICE site. Furthermore, the RICE AWS recorded also an average annual temperature of also -23.5°C. In contrast, the nearby Margaret AWS (located at 67m above sea level, just 96 km south-west of the RICE AWS, data obtained from Antarctic Meteorological Research Center and Automatic Weather Station Project; https://amrc.ssec.wisc.edu) records an average annual temperature of -26.6 °C. While recorded summer temperatures at RICE and Margaret AWS agree well, during winter the Margaret AWS records up to 10-15 °C colder 3-hourly temperatures than the RICE AWS. It is possible that rime build-up at the RICE AWS during winter (Supplementary Information, Figure S1) might have provided insulation that allowed for residual heat from the sensors to warm the temperature cavity leading to erroneously warm readings. Alternatively, it is possible that the Margaret AWS site is influenced by stronger temperature inversions leading to exceptionally cold temperatures of -60 °C, while the topography of Roosevelt Island might be less conducive to such conditions. A comparison between high resolution borehole temperature measurements conducted at RICE from November 2013 to November 2014 and AWS data, including snow temperature measurements, will provide important insights into this temperature off-set. Until this analysis is concluded, we argue that ERAi data, which agree well with the Margaret AWS observations, provide the most reliable temperature time series to calibrate the stable isotope – temperature relationship.

In Figure 2a, the ERAi surface air temperature (SAT) time series extracted for the RICE drill site is compared with the ERAi SAT spatial grid. The analysis suggests that temperature variability at RICE is representative of variability across the Ross Ice Shelf (with the exception of the western-most margin along the Transantarctic Mountains), Northern Victoria Land, western Marie Byrd Land, and the Ross and Amundsen Seas. Furthermore, the Antarctic Dipole pattern (Yuan and Martinson, 2001), a negative correlation between the SAT in the Ross / Amundsen Sea region and that of the Weddell Sea, is also captured in the data. The locations of the Siple Dome, WDC, Talos Dome, TALDICE, and Taylor Dome ice cores fall within the region of positive RICE SAT correlation.

The correlation between RICE $\delta D$ data and ERAi SAT (Figure 2b) retains a positive correlation across the Ross Ice Shelf and northern Ross Sea, but the WDC site now falls outside the field of the statistically significant correlation. The time series correlation between the ERAi SAT record and the RICE $\delta D$ data (Figure 2d) is r=0.42 (p<0.01). We test the robustness of this relationship by applying the minimum and maximum age solutions within the age uncertainty ($\leq 2$ year during this time period, Winstrup et al. 2017) to identify the age model solution within the age uncertainty that renders the highest correlation. This optimised solution RICE $\delta D_o$ is shown in Figure 2e with a correlation coefficient of r=0.66 (p<0.001). The correlation of the RICE $\delta D_o$ record with the ERAi SAT data (Figure 2c) produces a pattern that more closely resembles the correlation pattern using the ERAi data itself (Figure 2a), suggesting that the $\delta D$ record provides useful information about the regional temperature history.

From the comparison between RICE $\delta D_o$ and ERAi SAT records, we obtain a temporal slope of 3.37 ‰ °C$^{-1}$ (Figure 2f), which falls within the limit of previously reported values from Antarctica of ~2.90 – 3.43 ‰ °C$^{-1}$ for temporal (interannual) slopes (Schneider et al., 2005) and ~6.80 ± 0.57 ‰ °C$^{-1}$ for spatial slopes (Masson-Delmotte et al., 2008). We use this relationship to calculate temperature variations for the RICE $\delta D$ record. The average annual temperature calculated for 1979-2012 from ERAi for the RICE site is -27.4 ± 2.4 °C and for the $\delta D_o$ data: -27.5 ± 3.6 °C. Although the year to year SAT variability appears to be well captured in the RICE $\delta D_o$ record, there are discrepancies in observed trends. While RICE $\delta D_o$ data suggest an increase in SAT from 1996 onwards, ERAi SAT data do not show a trend (Figure 2e). It remains a challenge to determine how well reanalysis products, including ERAi data, and other observations, including isotope data, capture temperature trends in Antarctica (PAGES2k Consortium, 2013;Stenni et al., 2017) and thus whether the observed difference in the trend between ice core $\delta D_o$ and ERAi SAT is significant or meaningful. We also compare the $\delta D_o$ with records from AWS (Ferrell, Gill, and Margaret AWS) and stations (Byrd, McMurdo, Scott Base, Siple Stations) in the region (Figure S2 and Table S1, supplementary information). The comparison is hampered by the shortness of the records and gaps in the observations. Only years were used for which monthly values were reported for each month of the year. No statistically significant correlation was identified between $\delta D_o$ and available data.

Furthermore, we test the correlation with the near surface Antarctic temperature reconstruction by Nicolas and Bromwich (2014; referred to as NB2014), which uses three reanalysis products and takes advantage of the revised Byrd Station temperature record (Bromwich et al., 2013) to provide an improved reanalysis product for Antarctica for the time period 1958-2012 CE. We find no correlation between the NB2014 record and the ERAi data at the RICE site, nor the RICE $\delta D_o$ data for the 1979-2012 time period, perhaps suggesting some regional challenges. If the full time period available for the NB2014 data is considered (1957-2012), the NB2014 – RICE $\delta D_o$ correlation becomes weakly statistically significant with r=0.23 (p=0.09, Table S1).

## 4.2    Regional snow accumulation variability

Temporal and spatial variability of snow accumulation are assessed using 144 snow stakes covering a 200 km$^2$ array. The 3 m long, stainless steel poles were set and surveyed in November 2010, re-measured in January 2011, and revisited and extended

in January 2012 and November 2013. The measurements indicate a strong accumulation gradient with up to 32 cm water equivalent per year (weq a$^{-1}$) on the north-eastern flank decreasing to 9 cm weq a$^{-1}$ on the south-western flank. Near the drill site, annual average snow accumulation rates range from 22 ± 4 cm weq a$^{-1}$ from 2010 to 2013.

In addition, snow accumulation was measured using an ultrasonic sensor pounted on the RICE AWS. The sensor was positioned 140-160 cm above the ground and reset during each season. The 3-year record shows gaps (Figure 3) which represent times when rime on the sensor and/or blowing snow caused erroneous measurements. This condition was particularly prevalent during winter. Over the three years, the site received an average snowfall of ~75 cm a$^{-1}$. Assuming an average density of 0.37 g cm$^{-3}$ (average density from two snow pits, 0-75cm), the AWS data suggest ~20 cm weq a$^{-1}$ accumulation, which is comparable with the accumulation rate derived from the ice core (21 ± 6 cm weq a$^{-1}$ for the period 1979-2012, Winstrup et al. 2017). In contrast, ERAi data suggest an average annual snow accumulation of only 11 cm weq a$^{-1}$. It has been shown that Antarctic ERAi data capture precipitation variability, but generally underestimate the precipitation total (Wang et al., 2016;Sinclair et al.,2010;Bromwich et al., 2011). Further, the ERAi data are not directly comparable to local measurements because they do not capture periods of snow scouring by wind. Yet, there is a good agreement between the two data sets with respect to the timing and relative rate of precipitation, which suggests neither winters nor summers have been times of significant snow scouring. We attribute the difference between the ERAi data and our measurements to the spatial differences between measurements of the snow stake array, the interpolation field of the nearest ERAi data point, and the actual drill site location, as well as differences in assumed snow densities, or different methodologies in the conversion from precipitation to water equivalent units. The regional representativeness of RICE snow accumulation data is assessed by correlating the ERAi precipitation time series, extracted for the RICE location, with the ERAi precipitation grid data (Figure 4a). The correlation suggests that precipitation variability at Roosevelt Island is representative of the observed variability across the Ross Ice Shelf, the southern Ross Sea, and western West Antarctica. We note from Figure 4a that the sites of the Siple Dome ice core (green circle) and the West Antarctic Ice Sheet Divide ice core (WDC, red circle) are situated within the positive correlation field, while Talos Dome (purple circle) and Taylor Dome (orange circle) show no correlation to ERAi precipitation at RICE. A negative correlation is found with the region of the South Pacific, Antarctic Peninsula and eastern West Antarctica. In Figure 4b, the RICE snow accumulation record is correlated with ERAi precipitation data and shows similar pattern. The resemblance of the correlation patterns suggests that the variability of the RICE snow accumulation data (Figure 4b) reflects regional precipitation variability (Figure 4a) and thus likely can elucidate regional snow accumulation variability in the past, in particular for array reconstructions such as i.e. Thomas et al. (2017) . We also test the correlation with the optimised age scale derived for the $\delta D_o$ record and find that for snow accumulation data this adjusted age scale (Acc$_o$) reduces the correlation but remains statistically significant (Table 2). This result suggests that the optimised age solution is not superior to the RICE17 age scale and we note that the sensitivity of the correlation to those minor adjustments is founded in the brevity of the common time period. However, there is no significant difference between the overall pattern and relationships of the two age scale solutions.

## 4.3 Influence of regional sea ice variability on RICE isotope and snow accumulation

Sea ice variability has been shown to influence isotope values in precipitated snow, particularly in coastal locations (Küttel et al., 2012;Noone and Simmonds, 2004;Thomas and Bracegirdle, 2009) through the increased contribution of enriched water vapour during times of reduced sea ice and increased sensible heat flux due to a higher degree of atmospheric stratification leading to more vigorous moisture transport. Tuohy et al. (2015) demonstrated that for the period 2006-2012 ~40-60 % of precipitation arriving at Roosevelt Island came from local sources in the southern Ross Sea. In addition, Emanuelsson et al. (in review) demonstrated the important role of blocking events, that are associated with over 88 % of large precipitation events at RICE, on sea ice variability via meridional wind field anomalies.

Snow accumulation at RICE is negatively correlated with sea ice concentration (SIC) in the Ross Sea region and northern Amundsen Sea region (Figure 5a), which predominately represents sea ice exported from the Ross Sea. We observe that years of increased (decreased) SIC leading to reduced (increased) accumulation at RICE, confirming the sensitivity of moisture-bearing marine air mass intrusions to local ocean moisture sources and hence regional SIC. The correlation between ERAi SIC and the optimised RICE $\delta D_o$ record (Figure 5b) similarly shows a negative correlation of SIC in the Ross Sea (perhaps with the exception of the Ross and Terra Nova polynyas) and the northern Amundsen Sea suggesting more depleted (enriched) values during years of increased (reduced) SIC.

In Figure 5 c and d, the sea ice extent index ($SIE_J$) for the Ross-Amundsen Sea, developed by Jones et al. (2016), is correlated with ERAi SAT and precipitation data. The analysis also identifies the co-variance of SIE and SAT, with increasing (decreasing) sea ice coinciding with cooler (warmer) SAT. This suggests that during years of increased SIE, the Ross Ice Shelf and western Marie Byrd Land experience lower temperatures and less snow accumulation. The correlation between $SIE_J$ and ERAi precipitation at RICE is r= –0.67, and for $SIE_J$ and RICE snow accumulation r= –0.56 (Table 2). Moreover, the correlation between $SIE_J$ and ERAi SAT and $SIE_J$ and RICE $\delta D_o$ is also statistically significant with r= –0.38 and r= –0.58, respectively. The higher correlation with RICE $\delta D_o$ perhaps suggests that the influence of SIE in the Ross Sea region affects the RICE $\delta D$ record both through direct temperature changes in the region as well as fractionation processes that are independent of temperature, such as the lengthened distillation pathway to RICE during periods of more extensive SIE.

The ERAi SAT and ERAi precipitation data at RICE (Table 2) reveal a positive correlation over large areas of Antarctica with higher correlation coefficients over the eastern Ross Sea and eastern Weddell Sea (spatial fields not shown). At the RICE site, the correlation reaches r=0.66 (p<0.001). Moreover, the correlation between RICE $\delta D$ and RICE Acc [or RICE $\delta D_o$ and RICE $Acc_o$] data yield a statistically significant correlation of r=0.49 (p<0.01) [or $r_o$=0.62, $p_o$<0.001], respectively. This suggests that years with positive isotope anomalies are frequently characterised by higher snow accumulation rates. In contrast, precipitation during low snow accumulation years might be dominated by precipitation from air masses that have travelled further and perhaps across West Antarctica (Emanuelsson et al., in review) leading to more depleted isotope values and lower snow accumulation rates than local air masses from the Ross Sea region.

## 4.4    Influence of climate drivers on prevailing conditions in the Ross Sea region

Seasonal biases and the enhancing or compensating effects of the relative phasing of SAM, ENSO, and IPO conditions, on seasonal, annual and decadal time scales, make linear associations of climate conditions and their relationship with climate drivers in the South Pacific challenging. We use the $SAM_A$ index developed by Abram et al. (2014) to test the fidelity of the SAM relationship with the climatic conditions in the Ross Sea over the past millennium (Table 2). The $SAM_A$ is highly correlated (r=0.75) with the SAM record developed by Marshall (2003) for their common time period (1957-2009). In addition, the Southern Oscillation Index (SOI, Trenberth and Stepaniak, 2001), Niño 3.4 Index (Rayner et al., 2003b), Niño 4 Index (Trenberth and Stepaniak, 2001), and the IPO Index (Henley et al., 2015) are used to investigate the influence of SST variability in the eastern and central tropical Pacific on annual and decadal time scales (IPO). In addition, we take advantage of a 850-year reconstruction of the Niño 3.4 index (Emile-Geay et al., 2013) to investigate any long term influence of the eastern Pacific SST on environmental conditions in the Ross Sea.

The correlation of ERAi data and modern ice core records covering the 1979-2012 CE period with indices of relevant climate drivers (Table 2) suggests that $SAM_A$ has an enduring statistically significant relationship with temperature, snow accumulation and SIE in the Ross Sea, with the positive SAM being associated with cooler temperatures, lower snow accumulation / precipitation and more extensive SIE. The correlations remain robust and at comparable levels using a detrended $SAM_A$ record. In contrast, ENSO (SOI, Niño 3.4 and 4) and ENSO-like variability (IPO) have only linear statistically significant relationships with ERAi precipitation (but not with RICE snow accumulation) and $SIE_J$. The dynamic relationship

between the phasing of SAM, ENSO, and IPO maybe masking aspects of the interactions (Fogt and Bromwich, 2006; Marshall and Thompson, 2016). In 2010, an anomalously cold year is observed. If only the time series from 1979-2009 is considered, the correlations between RICE $\delta$D and these considered climate drivers becomes statistically significant: SOI (r=-0.48, p=0.006), Niño 3.4 (r=0.48, p=0.007); Niño 4 (r=0.57, p=0.001), and IPO (r=0.44, p=0.014). This further highlights the vulnerability of this analysis to individual years due to the brevity of the time series further complicating a linear analysis between individual drivers and regional responses. Moreover, statistically significant correlations might also be obtained if seasonal averages could be used for the comparison as ENSO events usually peak during the austral summer, in particular December (Turner, 2004). Nonetheless, this analysis confirms previous findings that the Ross Sea Region is sensitive to the cumulative, independent and dependent influences of SAM, ENSO and the IPO.

## 5    Temperature and snow accumulation variability over the past 2.7 ka

Decadally-smoothed RICE isotope and snow accumulation records for the past 2.7 ka (Figure 6) are compared with published data from the Ross Sea region (Siple Dome – green), coastal East Antarctica (Talos Dome / TALDICE – purple, Taylor Dome - orange) and West Antarctica (West Antarctic Ice Sheet Divide Ice Core / WDC - red).

### 5.1    Regional Temperature Variability

We find that the RICE and Siple Dome (Brook et al., 2005) water isotope records share a long-term isotopic warming trend in the Ross Sea Region. In contrast, WDC isotope (Steig et al., 2013) and borehole temperature data (Orsi et al., 2012) exhibit a long-term cooling trend for West Antarctica, while TALDICE (Stenni et al. 2012) and Taylor Dome (Steig et al. 1998, Steig et al. 2000) recorded stable isotopic conditions for coastal East Antarctica in the western Ross Sea.

Elevation changes influence water isotope values (Vinther et al., 2009). Thinning of Roosevelt Island, inferred from the amplitude of arched isochrones beneath the crest of the divide and the depth-age relationship from the ice core (Conway et al., 1999) is less than 2 cm a$^{-1}$ for the past 3.5 ka. Thus, the surface elevation has decreased less than 50 m over the past 2.7 ka. Assuming that these elevation changes are sufficient to influence vertical movement of the precipitating air mass, such an elevation change could account for an isotopic enrichment of $\delta$D = ~2 ‰ (Vinther et al., 2009), which is insufficient to explain the total observed increase of 8 ‰. Furthermore, the RICE $\delta$D trend is characterised by two step-changes at 580 CE $\pm$ 27 years and 1477 CE $\pm$ 10 years (grey dotted lines in Figure 6). These change points were identified using minimum a threshold parameterisation to achieve a minimised residual error (Killick et al., 2012; Lavielle, 2005). At the identified change points, the decadal isotope values increase by ~3 ‰ and ~5 ‰, respectively, which suggests that elevation changes were not a principal driver. Using the temporal slope of 3.37 ‰ per °C, these abrupt temperature transitions could represent an increase of the average decadal temperature (Figure 7a) from -28.5 °C to -27.7 °C and from -27.7 °C to -26.2 °C, respectively. An underlying influence from long-term thinning, accounting for a $\delta$D shift of 2.4‰, would exaggerate the observed warming of 2.3 °C by 0.7 °C, thus suggesting a minimum isotopic temperature warming of at least 1.6 °C. The modern decadal isotope temperature average (2003-2012) of –25.1 °C (pink bar, Figure 7a) and ERAi temperature of -26.5 °C lie within the 1 σ distribution of the natural decadal temperature variability of the past 500 years. We note that changes in atmospheric circulation and sea ice extent might also have contributed significantly to the change in the observed isotopic shift and we offer some suggestions in the following sections.

The Siple Dome ice core $\delta^{18}$O record exhibits a similar isotope history to the RICE $\delta$D record. Siple Dome isotope data reveal an abrupt warming or isotopic enrichment at 605 CE, some 25 years later than in RICE, but within the cumulative age uncertainty of the two records. After 605 CE, Siple Dome isotope temperatures remain stable, although recording somewhat

warmer isotope temperatures from about 1875 CE. Late-Holocene elevation changes (thinning) at Siple Dome have been reported to be negligible (Price et al., 2007) and are unlikely to have caused the observed abrupt isotopic shift at 605 CE. In contrast to records from the western Ross Sea (Bertler et al., 2011;Rhodes et al., 2012;Stenni et al., 2002) and West Antarctica (Orsi et al., 2012), RICE and Siple Dome do not show an isotopic warming or cooling associated with the Medieval Warm Period (MWP) or the Little Ice Age (LIA), respectively.

The WDC $\delta^{18}O$ record suggests a long-term isotope cooling of West Antarctica, confirmed by borehole temperature reconstructions (Orsi et al., 2012). This trend is consistent with warmer-than-average temperatures during the MWP and cooler conditions during the LIA, but may also reflect changes in elevation and decreasing insolation (Steig et al., 2013). The cooling trend is followed by an increase in temperature in recent decades (Orsi et al., 2012;Steig et al., 2009) consistent with an increase in marine air mass intrusions (Steig et al., 2013). We note that within 200 years of the onset of the isotopic warming at RICE (at 580 CE ± 27 years), the WDC borehole temperature and isotope data start to record a temperature decline, in line with the observed anti-phased relationship of WDC with RICE and Siple Dome. No notable change is observed in WDC water stable-isotope temperature data in the late 15th Century. In contrast, WDC borehole temperature suggests the onset of a warming trend within the last 100 years, marking the modern divergence between WDC isotope and borehole records. The TALDICE and Taylor Dome water stable-isotope temperatures do not exhibit a long term trend over the past 2.7 ka. Yet colder water stable-isotope temperature anomalies have been associated with the LIA period at both sites (Stenni et al., 2002), which coincide with cooler conditions at WDC and the intensified increase in isotope temperature at RICE.

The similarity between the RICE and Siple Dome isotope records suggests that the eastern Ross Sea was dominated by regionally-coherent climate drivers over the past 2.7 ka, perhaps receiving precipitation via similar air-mass trajectories. Overall this comparison shows that isotope temperature trends in the eastern Ross Sea (isotopic warming at RICE and Siple Dome) and West Antarctica (WDC cooling) were anti-phased for over 2 ka (660 BCE to ~1500 CE), while the western Ross Sea (TALDICE) remained stable, forming a distinct Ross Sea Dipole pattern. From the 17th Century onwards this relationship changes. While WDC water stable isotope temperatures continue to cool, from the 17th Century, the WDC borehole temperature records a warming. At the same time, RICE and Siple Dome experience warmer isotope temperatures while TALDICE recovers from its coldest recorded isotope temperature during the study period.

## 5.2 Regional snow accumulation variability

Investigating long-term trends in snow accumulation records (Figure 6b), the decadally-smoothed RICE show a discernible positive trend from about 600 CE of about 0.2±0.1 cm weq per century (Winstrup et al. 2017) for the eastern Ross Sea, while WDC (Fudge et al., 2016) displays a decreasing trend for central West Antarctica. The RICE snow accumulation data reach a maximum in the 13th Century, with a trend towards lower values from the late 17th Century onward of about -0.9±0.6 cm weq per century (Winstrup et al. 2017). Since 1950 CE, this trend accelerated with a decreasing rate of -6.6 cm weq per century (Winstrup et al. 2017). Based on the negative correlation between RICE snow accumulation and SIC in recent decades, we interpret the long-term increase in snow accumulation to represent a long term reduction in SIC the Ross Sea, consistent with a long term increase in RICE isotope temperature. The recent rapid decline in snow accumulation rates could be related to increases in sea ice conditions in the eastern Ross Sea, perhaps marking the modern onset of local sea ice expansion. The modern decadal average (2002-2012) of 20 cm weq a$^{-1}$ lies within the 2 σ variability of decadal RICE ice core snow accumulation rates (Figure 7). Talos Dome records its maximum snow accumulation rates at the end of the 13th Century and reduced snow accumulation during the Little Ice Age period (Stenni et al., 2002). Until the 15th Century, RICE and WDC snow accumulation and isotope data show an expected positive correlation between their respective isotope and snow accumulation records. Regionally, RICE and WDC isotope and snow accumulation are anti-phased. From the 16th Century, this relationship reverses and RICE and WDC snow accumulation are now in phase, suggesting below average snow accumulation in the eastern

Ross Sea and West Antarctica. We note that from the 16th Century snow accumulation at RICE and WDC display a negative correlation with water stable isotope (RICE) and borehole temperature (WDC) reconstructions, respectively. At RICE this relationship is again positive for 1979-2012. During the 20th Century, Talos Dome records an increase in snow accumulation of ~11% in the western Ross Sea (Stenni et al., 2002), in contrast to observed snow accumulation decrease at RICE and WDC. However, the values are not unique within the variability of the 800 years captured in the Talos Dome record.

## 6 Drivers and patterns of decadal to centennial climate variability

Paleo-reconstructions of the SAM$_A$ (Abram et al., 2014) and Niño 3.4 (Emile-Geay et al., 2013) indices (Figure 6) provide important opportunities to investigate the influence of dominant drivers of regional climate conditions over the past millennium. The Niño 3.4 index captures in particular the ENSO signal originating in the central-eastern tropical Pacific associated with the PSA1 pattern. Emile-Geay and colleagues (2013) note that while the reconstructions based on three model outputs agree well on decadal to multidecadal time periods, they show different sensitivities at centennial resolution (but not the sign). The SAM index (Thompson and Wallace, 2000;Thompson and Solomon, 2002a) was developed during the late 20th Century, at a time when SAM was characterised by a strong positive trend. The SAM$_A$ reconstruction showed that the modern SAM is now at its most positive state of the past millennium (Abram et al., 2014). As a consequence, the reconstructed SAM$_A$ index is mainly negative. To investigate the influence of positive and negative anomalies of the SAM$_A$ relative to its average state over the past 1 ka, we plot the SAM$_A$ reconstruction with an above (light purple) / below (purple) value of its long term average of -1.3 (instead of '0'). The traditional positive SAM$_A$ values (above 0, dark purple) are also shown for reference. Assessing the relationship between RICE, Siple Dome, WDC and TALDICE, we identify three major time periods of change: 660 BCE to 1367 CE (long-term baseline), 1368 to 1683 CE (negative SAM$_A$) and 1684 to 2012 CE (onset of the positive SAM$_A$).

### 6.1 Long-term baseline 660 BCE to 1367 CE

We find that for over 2 ka – from $660 \pm 44$ years BCE to $1367 \pm 12$ CE years, the eastern Ross Sea (RICE and Siple Dome) shows an enduring antiphase relationship with West Antarctica (WDC), while coastal East Antarctica in the western Ross Sea (TALDICE, Taylor Dome) remains neutral (Figure 6). Moreover, with some minor exceptions, isotope and snow accumulation records at RICE, WDC, and TALDICE, respectively, are positively correlated.

### 6.2 Negative SAM$_A$ - 1368 CE to 1683 CE

The SAM$_A$ reconstruction suggests, that over the past millennium, the SAM was at its most negative (Abram et al., 2014) from $1368 \pm 12$ years CE to $1683 \pm 8$ CE years. As noted by Abram et al. (2014), the SAM$_A$ and Niño 3.4 reconstructions (Emile-Geay et al., 2013) are anti-phased on multi-decadal to centennial time scales with the Niño 3.4 index recording some of the warmest SSTs over the past 850 years during this period of negative SAM$_A$.

During the negative SAM$_A$, RICE shows a distinct and sudden increase in isotope temperature, while TALDICE records its coldest conditions over the past 2.7ka. The Taylor Dome record also shows prolonged cold isotope temperature anomalies during this time period. Previously published shorter records from the western Ross Sea from Victoria Lower Glacier in the McMurdo Dry Valleys (Bertler et al., 2011) and Mt Erebus Saddle (Rhodes et al., 2012) also suggest colder conditions in the western Ross Sea during this period, with more extensive sea ice and stronger katabatic flow. We observe that WDC and TALDICE show below average snow accumulation values, while RICE snow accumulation changes from a long-term positive to a negative trend. Such trends are consistent with the reported increased SIE in the western Ross Sea (colder SAT, lower

snow accumulation, at TALDICE; and cooler conditions with more extensive sea ice and stronger katabatic winds at Victoria Lower Glacier and Mt Erebus) and Bellingshausen Sea (less snow accumulation, colder SAT at WDC). In contrast, RICE records warmer isotope temperatures along with less and more variable snow accumulation, displaying a distinct Ross Sea Dipole. Coinciding with the sudden increase in RICE $\delta$D in 1492 CE is the decoupling of the local isotope temperature from snow accumulation, evident from the diversion of the RICE snow accumulation and $\delta$D trends. The reduction in snow accumulation might be linked to a negative SAM-induced weakening of the ASL, perhaps leading to the development of fewer blocking events in the eastern Ross Sea. Alternatively, the abrupt change to warmer isotope temperatures at RICE might also point towards the development of the Roosevelt Island polynya. In recent decades, a Roosevelt Island polynya is observed and merges at times with the much larger Ross Sea polynya (Morales Maqueda et al., 2004). In contrast to the Ross Sea Polynya (Sinclair et al., 2010), a local polynya could provide a potent source for isotopically enriched vapour to precipitation at RICE, perhaps exaggerating the actual warming of the area as interpreted from water stable isotope data. We expect the influence of a Roosevelt Island polynya to have a reduced effect on the more distant Siple Dome, which is consistent with our observations.

## 6.3 Onset of the positive SAM - 1684 CE to 2012 CE

At 1684 CE $\pm$ 7 years, the SAM$_A$ increases and remains above its long-term average until modern times while the Niño 3.4 index suggests a change to the prevalence of strong La Niña-like conditions, conditions conducive to a strengthening of the ASL. RICE $\delta$D suggest the continuation of warm, isotopically enriched conditions, while snow accumulation drops below the long-term average, with the RICE snow accumulation trend now in-phase with the negative trend at WDC (Figure 6). At the same time, Talos Dome records highly variable snow accumulation rates with an 11 % increase in snow accumulation rates from the 20$^{th}$ Century onwards. The change to above-average SAM$_A$ (or even positive SAM) values coincides with the onset of the diversion of WDC water isotope and borehole temperature reconstructions and the decoupling of the RICE $\delta$D and snow accumulation trends. Coincident positive SAM (purple SAM$_A$ values in Figure 6c) and La Niña events have been linked in recent decades to increases in SIE in the western Ross Sea and decreases in the Bellingshausen Sea (Stammerjohn et al., 2008). This is consistent with the notable reduction in snow accumulation at RICE and the trend towards warmer conditions and increased marine air mass intrusions into West Antarctica as inferred from WDC isotope data (Steig et al., 2013) but is inconsistent with the reduction in snow precipitation at WDC and the trend to warmer isotope temperatures at RICE. We interpret the continuation of warm RICE isotope temperatures to reflect the persistence of the Roosevelt Island polynya.

## 6.4 Dipole pattern on decadal to centennial time scales

To investigate the drivers of decadal to centennial variability, we compare the linearly detrended time series of RICE water stable isotope records with those from (i) Siple Dome and WDC (West Antarctica, Figure 8a) and (ii) TALDICE (East Antarctica, western Ross Sea, Figure 8b). The analysis suggests that from about 500 CE onwards, RICE and Siple Dome variability are in-phase with West Antarctic climate variability (WDC). In contrast, RICE and TALDICE isotope variability alternates between spatial pattern of in-phase (purple) and anti-phase (grey, Ross Sea Dipole) relationships. During the negative phase of the SAM$_A$ (Figure 8), the anti-correlation between RICE and TALDICE is particularly strong.

To assess the correlation of cyclicities apparent in the RICE, Siple Dome, WDC and TALDICE isotope records, wavelet coherence spectrum analyses were conducted (Figure 9) on the time series shown in Figure 8. The analysis of RICE and Siple Dome (Figure 9a) suggests that the two records positively correlate at a broad spectrum of frequencies with cyclicities between 200 – 500 years. The correlation is weakest during 660-100 BCE. A strong anti-phased coherence is also observed for the 30-70 year periodicity from 100-800 CE. The high coherency suggests that RICE and Siple Dome respond to similar forcings. The coherence analysis between RICE and WDC shows an enduring in-phase correlations from ~1000 CE to today for the bandwidth of 200 - 700 years. An anti-phase coherence is found from 0 - 500 CE. The coherence analysis between RICE and

TALDICE identifies strong relationships predominantly for the early part of the records, from 660 BCE to ~800 CE and a weak coherence from about 800-1700 CE, when RICE leads by ~75-100 years. The analysis suggests that for the past 2.7 ka the eastern Ross Sea (RICE, Siple Dome) and western West Antarctica (WDC) are climatologically closely linked in their response to forcings on decadal to centennial time scales and are positively correlated for the past 1.6 ka. In contrast, the relationship between the western (TALDICE) and eastern (RICE, Siple Dome) Ross Sea is more variable.

## 7       Concluding Remarks

The recent change to a strongly negative SAM (Marshall Index -3.12) in November 2016 coincided with a significant reduction of Antarctic SIE, including the Ross Sea, during the 2016/17 summer (Turner et al., 2017). Longer observations are necessary to assess whether this recent trend continues and indeed forces the reduced SIE, but it fuels questions on the potential acceleration of future environmental change in the Antarctic / Southern Ocean region. To improve projections for the coming decades, an improved understanding of the interplay of teleconnections and local feedbacks is needed.

The Ross Sea region is a climatologically sensitive region that is exposed to tropical and mid-latitude climate drivers. The ASL has shown to deepen during coinciding positive SAM and La Niña events, and to weaken during negative SAM and El Niño events. Such interactions have far reaching implications on the regional atmospheric and ocean circulations and sea ice (Turner et al., 2015, Raphael et al., 2016). Additionally, a negative (positive) IPO leads to cooler (warmer) SSTs in the Ross, Amundsen and Bellingshausen Seas and has the potential to strengthen (in phase) or weaken (out of phase) the ENSO teleconnection (Henley et al., 2015). Furthermore, the phasing and strength of ENSO events and SAM have been shown to be time dependent (Fogt and Bromwich, 2006).

Our data suggest that changes in these dynamically linked climate patterns coincide with significant and abrupt changes in the past with implications for regional interpretations of trends, including temperature, mass balance and SIE. For over 2 ka, from 660 BCE to the late 14th Century, climate trends in the eastern Ross Sea (RICE and Siple Dome, trend to warmer isotope temperatures and higher precipitation) are anti-correlated with conditions in the western West Antarctica (WDC, isotopic cooling with reduced precipitation), while coastal East Antarctica in the western Ross Sea appeared decoupled (TALDICE/Taylor Dome and Talos Dome, no trend in isotope temperature or precipitation, respectively). This regional pattern re-organised during a period with strongly negative SAM conditions (SAM$_A$) accompanied by exceptionally warm tropical SST (Niño 3.4 index). In modern times, such conditions cause a weakening of the ASL which in turn can lead to a reduction of marine air mass intrusions into West Antarctica. Indeed, we observe that western West Antarctica (WDC borehole and isotope temperature) and the western Ross Sea (TALDICE) show cold temperatures during this time period that coincides with the Little Ice Age, while the eastern Ross Sea (RICE, Siple Dome) shows warmer or stable isotope temperatures, respectively. In the late 17th Century, the SAM$_A$ changes to above average values, concurrent with a change to strong La Niña-like conditions, a framework conducive to a deepening of the ASL. Now, West Antarctica (WDC borehole temperature), the eastern Ross Sea (RICE, Siple Dome) and the western Ross Sea (TALDICE) all experience warmer isotope temperatures. At the same time, however, we observe reduced snow accumulation at RICE and WDC and an increase at Talos Dome. We interpret this pattern to reflect an increase in SIE in the eastern Ross Sea with perhaps the establishment of the modern Roosevelt Island polynya as a local moisture source for RICE. The continued improvements of array reconstructions (Stenni et al., 2017; Thomas et al., 2017) and the assessment of isotope-enabled climate models are an exciting development to further our knowledge of the drivers and effects of past change and their implications for future projections.

Data availability.

The following new RICE data are made available in this manuscript:

- RICE water stable isotopes (δD) have been archived at the PANGAEA data base: https://doi.pangaea.de/10.1594/PANGAEA.880396
- GPS and Radar data have been archived at the U.S. Antarctic Program Data Center: https://gcmd.gsfc.nasa.gov/search/Metadata.do?entry=USAP-0944307&subset=GCMD

The following published RICE data used in this manuscript are accessible via:

- RICE17 age scale has been archived at the PANGAEA data base: https://doi.pangaea.de/10.1594/PANGAEA.882202
- RICE snow accumulation data have been archived at the PANGAEA data base: https://doi.pangaea.de/10.1594/PANGAEA.882202

Sources of published ice core data used in this manuscript:

- WDC water stable isotopes (δ$^{18}$O) have been accessed via: https://www-nature-com.helicon.vuw.ac.nz/articles/nature12376#supplementary-information
- WDC snow accumulation data have been accessed via: https://www-nature-com.helicon.vuw.ac.nz/articles/nature12376#supplementary-information
- WDC borehole temperature has been accessed via: https://nsidc.org/data/NSIDC-0638/versions/1

- Talos Dome snow accumulation data have been accessed via: https://www.ncdc.noaa.gov/paleo/study/22712
- TALDICE water stable isotope data (δ$^{18}$O) have been accessed via: https://www1.ncdc.noaa.gov/pub/data/paleo/pages2k/stenni2017antarctica/
- Siple Dome water stable isotopes (δ$^{18}$O) have been accessed via: https://www-nature-com.helicon.vuw.ac.nz/articles/nature12376#supplementary-information

- Taylor Dome water stable isotopes (δ$^{18}$O) have been accessed via: https://www1.ncdc.noaa.gov/pub/data/paleo/pages2k/pages2k-temperature-v2-2017/data-current-version/Ant-TaylorDome.Steig.2000.txt

Sources of meteorological data and climate indices used in this manuscript:

- SAM$_A$ Index developed by Abram et al. 2014 has been accessed via
ftp://ftp.ncdc.noaa.gov/pub/data/paleo/contributions_by_author/abram2014/abram2014sam.txt
- Niño 3.4 Index developed by Emile-Geay et al. (2012) has been accessed via ftp://ftp.ncdc.noaa.gov/pub/data/paleo/contributions_by_author/emile-geay2012/emile-geay2012.xls
- Niño 3.4 Index (Rayner et al., 2003a) has been accessed via http://www.esrl.noaa.gov/psd/gcos_wgsp/Timeseries/Nino34/
- Niño 4 Index (Rayner et al., 2003a) has been accessed via https://www.esrl.noaa.gov/psd/gcos_wgsp/Timeseries/Nino4/
- SOI (Allan et al., 1991) has been accessed via http://www.cru.uea.ac.uk/cru/data/soi/
- Ross/Amundsen Sea sea ice extent data (SIEJ) developed by Jones et al. (2016) have been accessed via http://www.nature.com/articles/nclimate3103#supplementary-information
- Byrd Station meteorological data (Bromwich et al., 2013) have been accessed via the Byrd Polar Research Centre, Polar Meteorological Group, Ohio State University http://www.polarmet.osu.edu/datasets/Byrd_recon/
- Meteorological data for Ferrell, Gill, and Margaret AWS have been accessed via Antarctic Meteorological Research Center and Automatic Weather Station Project https://amrc.ssec.wisc.edu
- Data for McMurdo Station and Scott Base are accessed via the MET-READER
https://legacy.bas.ac.uk/met/READER/data.html
- IPO Index (Henley et al., 2015) has been accessed via http://www.esrl.noaa.gov/psd/data/timeseries/IPOTPI/

- NB2014 - near-surface Antarctic temperature reconstruction data (NB2014, Nicolas and Bromwich, 2014) have been accessed via http://polarmet.osu.edu/datasets/Antarctic_recon/

Supplementary information.

Competing interests.

The authors declare that they have no conflict of interest.

**Acknowledgments.**

Funding for this project was provided by the New Zealand Ministry of Business, Innovation, and Employment Grants through Victoria University of Wellington (RDF-VUW-1103, 15-VUW-131) and GNS Science (540GCT32, 540GCT12), and Antarctica New Zealand (K049), the US National Science Foundation (US NSF ANT-0944021, ANT-0944307, ANT-1443472), British Antarctic Survey Funding (BAS PSPE), the Center of Ice and Climate at the Niels Bohr Institute through

the Carlsberg Foundation's "North-South Climate Connection" project grant, and the Major State Basic Research Development Program of China (Grant No. 2013CBA01804). We are indebted to Hedley Berge, Jeff Rawson, Margie Grant, Lou Albershardt, and Antarctica New Zealand staff at Scott Base and in Christchurch for their support of the RICE field seasons. We are grateful for the support by US 109th New York Air National Guard (NYANG) LC-130 Hercules and Canadian Kenn Borek aircraft crews for their excellent support into and out of Roosevelt Island. Furthermore, we would like to thank Stephen

Mawdesley, Grant Kellett, Ryan Davidson, Ed Hutchinson, Bruce Crothers and John Futter of the Mechanical and Electronic Workshops of GNS Science for technical support for the international RICE core progressing campaigns. We would like to thank Beaudette Ross for conducting gas isotope measurements at the Scripps Institution of Oceanography, University of California, San Diego. We are grateful to Diane Bradshaw and Bevan Hunter for their assistance in naming the New Zealand ice core drill. We would like to thank Barbara Stenni for advice and discussions on the Talos Dome and TALDICE records.

We thank two anonymous reviewers to help us make important improvements to this manuscript. This work is a contribution to the Roosevelt Island Climate Evolution (RICE) Program, funded by national contributions from New Zealand, Australia, Denmark, Germany, Italy, the People's Republic of China, Sweden, UK, and USA. Logistics support was provided by Antarctica New Zealand (K049) and the US Antarctic Program.

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

**Table 1: Overview of ice core records used in this manuscript. Locations are present in Figure 1.**

| Name | Location | | Elevation (m) | Drill Depth (m) | Age Scale | Year Recovered | Reference |
|---|---|---|---|---|---|---|---|
| **RICE Deep** | S 79.3640 | W 161.706 | 550 | 8.57-764.60 | RICE17 | 2011/12 (0-130m) and 2012/13 (130-764.60m) | This paper |
| **RICE 12/13 B** | S 79.3621 | W 161.700 | 550 | 0-19.41 | RICE17 | 2012/13 | This paper |
| **WDC** | S 79.47 | W 112.09 | 1,766 | 3,405 | WDC06A-7 | 2006-11 | Steig et al. 2013, Fudge et al. 2016 |
| **Siple Dome** | S 81.65 | W 148.81 | 620 | 1,004 | Brook et al.2005 | 1997-99 | Brook et al., 2005 |
| **TALDICE** | S 72.78 | E 159.07 | 2,318 | 1,620 | Severi et al. 2012 | 2005-07 | Stenni et al. 2011, Buiron et al. 2011, Severi et al. 2012 |
| **Talos Dome (TD96)** | S 72.80 | E 159.06 | 2,316 | 89 | Stenni et al. 2002 | 1995 | Stenni et al. 2002 |
| **Taylor Dome** | S 77.70 | E 159.07 | 2,375 | 554 | WDC06A-7 | 1993-1994 | Steig et al.1998, Steig et al. 2000; Sigl et al. 2014, PAGES2k Consortium 2017 |

**Table 2:** Overview of correlation coefficients for annual means of the common time period 1979-2012 between climate parameters, proxies and indices: the original RICE ($\delta D$) and optimised ($\delta D_o$) data (this paper), the original RICE snow accumulation data (RICE Acc, Winstrup et al., 2017) and data adjusted to the revised age scale of $\delta D_o$ – $Acc_o$, ERAi Surface Temperature (ERAi SAT) and Precipitation (ERAi Precip), Dee at el., 2011), Ross/Amundsen Sea Sea Ice Extent ($SIE_J$, Jones et al., 2016), Southern Annular Mode Index ($SAM_A$, Abram et al., 2014), Southern Oscillation Index (SOI, Trenberth and Stepaniak, 2001), Niño 4 Index (Trenberth and Stepaniak, 2001) and Niño 3.4 (Emile-Geay et al., 2013), Inter-decadal Pacific Oscillation Index (IPO, Henley et al.,2015), and the near-surface Antarctic temperature reconstruction (NB2014, Nicolas and Bromwich, 2014). Significance values are adjusted for degree of freedom depending on the length of the time series. Only correlation coefficients exceeding 95% ($r \geq 0.34$, n=34) are shown; bold-italic values exceed 99% ($r \geq 0.42$, n=34); bold values exceed 99.9% ($r \geq 0.54$, n=34). $SAM_A$ and IPO have been adjusted for a lower degree of freedom (df=28) as the reconstructions end in 2007. Nss denotes 'not statistically significant'. Correlation between RICE $\delta D$ and RICE Acc is r=0.49, p<0.01; RICE $\delta D_o$ and RICE $Acc_o$ is r=0.62, p<0.01.

| R | ERAi SAT | ERAi Precip | $SIE_J$ | $SAM_A$ | SOI | Niño 4 | Niño 3.4 | IPO | NB2014 |
|---|---|---|---|---|---|---|---|---|---|
| **RICE $\delta D/\delta D_o$** | *0.42/0.66* | *0.36/0.43* | *-0.49/-0.58* | nss/-0.40 | nss/nss | nss/nss | nss/nss | nss/nss | nss/nss |
| **RICE $Acc/Acc_o$** | **0.60**/0.39 | **0.67**/0.42 | **-0.56**/*-0.44* | *-0.46*/nss | nss/nss | nss/nss | nss/nss | nss/nss | nss/nss |
| **ERAi SAT** | x | **0.66** | -0.38 | *-0.49* | nss | nss | nss | Nss | nss |
| **ERAi Precip** | **0.66** | x | **-0.67** | *-0.42* | *-0.49* | 0.37 | 0.39 | *0.44* | nss |
| **$SIE_J$** | -0.38 | **-0.67** | x | *0.45* | **0.55** | *-0.48* | *-0.48* | **-0.58** | nss |

**Figures**

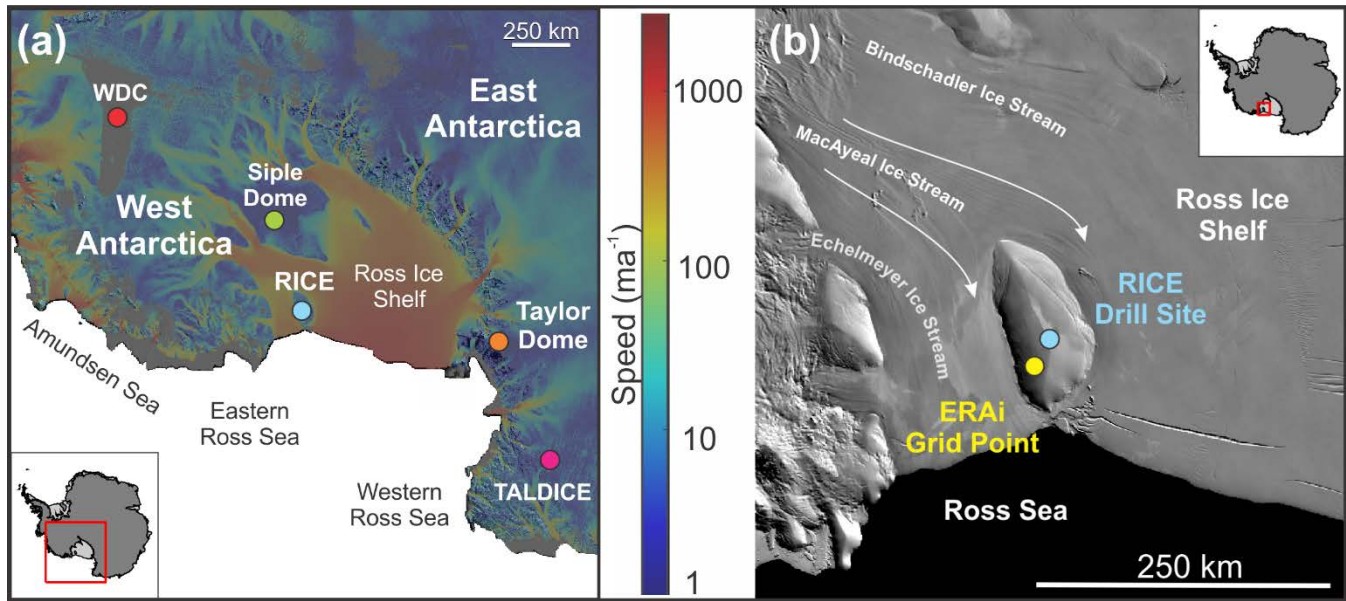

Figure 1: (a) Overview map of the Ross Sea region and eastern West Antarctica. Antarctic ice velocity derived from ALOS PALSAR, Envisat ASAR, RADARSAT- 2 and ERS-1/2 satellite radar interferometry colour coded on a logarithmic scale (Rignot et al., 2011). Coloured dots indicate the locations of ice core drilling sites used in this manuscript: RICE (blue), WDC (red), Siple Dome (green), TALDICE/Talos Dome TD96 (purple), Taylor Dome (orange); (b) Overview map of Roosevelt Island derived from Modis satellite images (Scambos et al., 2007). The maps were created using the Antarctic Mapping Tool (Greene et al., 2017). The coloured dots indicate the location of the RICE drill site (blue) and the nearest ERAi grid point (yellow).

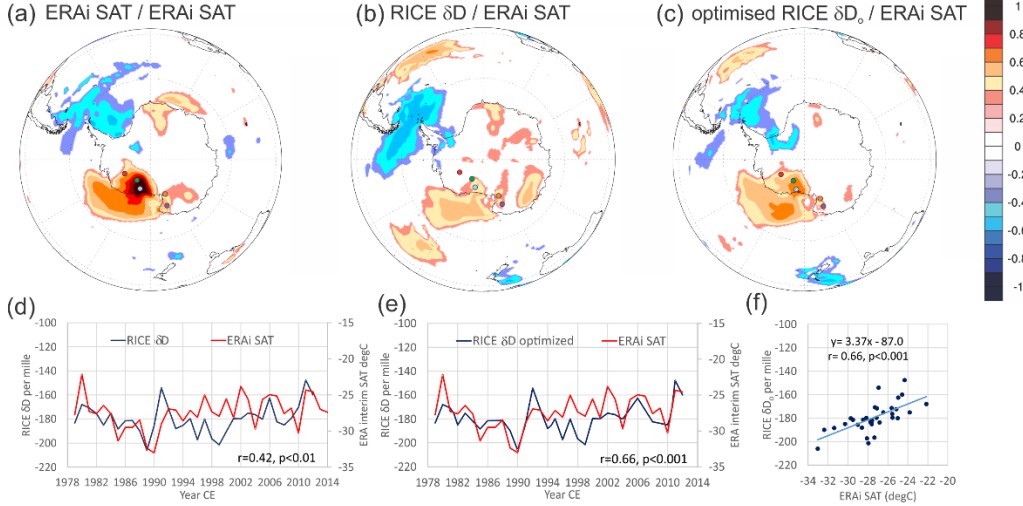

Figure 2: Spatial correlation fields exceeding ≥ 95% significance between a) ERAi annual SAT at the RICE site with ERAi annual SAT in the Antarctic / Southern Ocean region and b) ERAi annual SAT and annually averaged RICE δD data, c) as for b but with optimised RICE δD data alignment within the dating uncertainty. The correlation has been performed using ClimateReanalyzer.Org, University of Maine, USA. Comparison of the ERAi SAT time series with d) RICE δD data and e) optimised RICE δD data alignment. Panel f) scatter plot between RICE $\delta D_o$ and ERAi SAT. The coloured dots indicate the locations of the drill sites – RICE (blue), Siple Dome (green), WDC (red), TALDICE (pink), and Taylor Dome (orange).

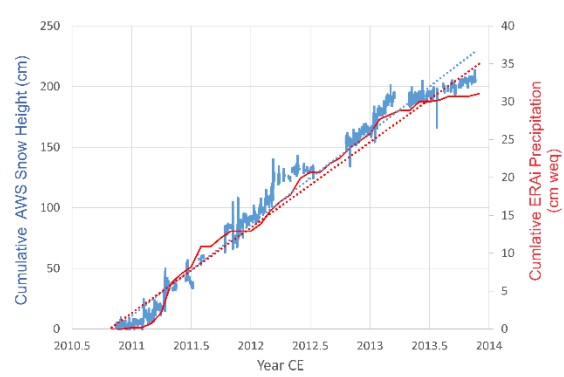

**Figure 3: Comparison of snow accumulation data recorded by the RICE AWS (blue) in cumulative snow height in cm and ERAi precipitation values (red) in cm water equivalent cumulative height for the RICE Drill Location. Dotted blue and red lines indicate linear trends for AWS and ERAi data, respectively.**

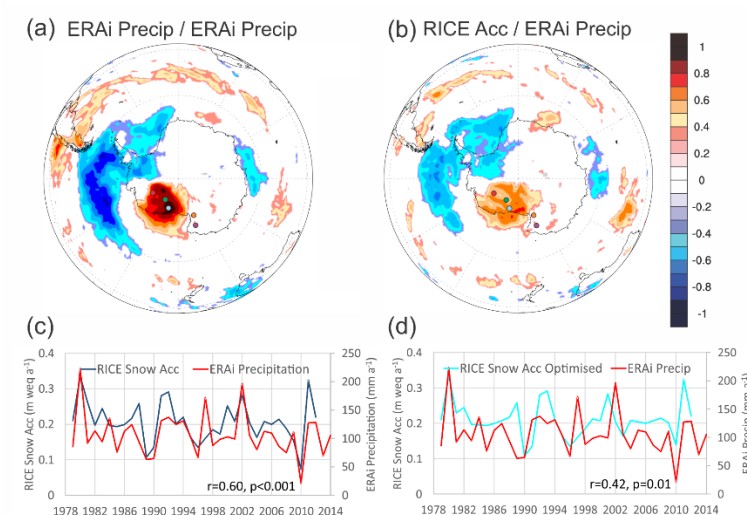

**Figure 4: a) Spatial correlation between ERAi annual precipitation at the RICE site with ERAi annual precipitation in the Antarctic / Southern Ocean region and b) spatial correlation between ERAi annual precipitation and annually averaged RICE snow**
10 **accumulation data. Only fields exceeding ≥ 95% significance are shown. The correlation has been performed using ClimateReanalyzer.Org, University of Maine, USA. Comparison of the ERAi precipitation time series with d) RICE snow accumulation data and e) RICE snow accumulation with optimised RICE δD data alignment. Coloured dots indicate locations of WDC (red), Siple Dome (green), RICE (blue), TALDICE/Talos Dome (purple), and Taylor Dome (orange).**

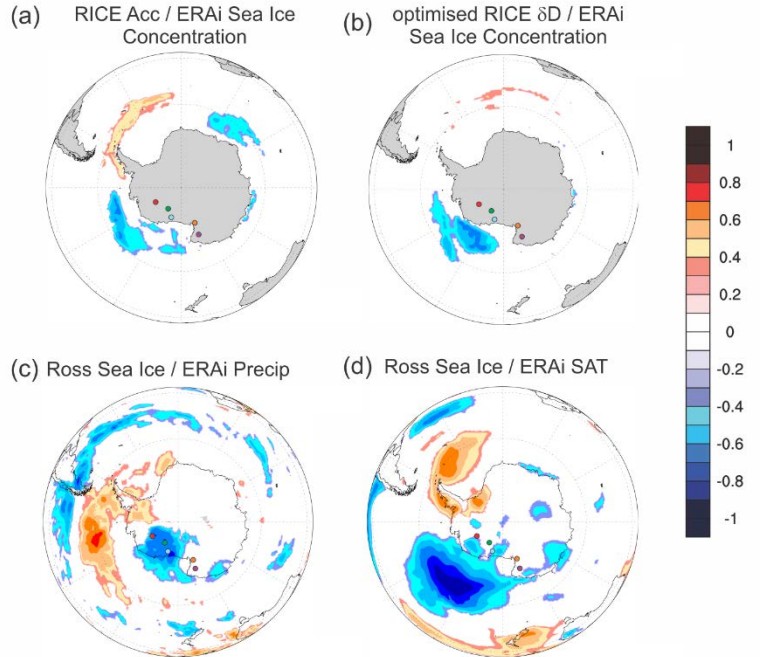

5 **Figure 5: Upper panels: Spatial correlation of ERAi sea ice concentration (SIC) fields with the time series of a) RICE snow accumulation and b) RICE δDo. Lower panels: spatial correlation of the Ross-Amundsen Sea Sea Ice Extent (SIE_J) time series (Jones et al., 2016) with c) ERAi Precipitation and d) ERAi SAT fields. Only fields exceeding ≥95% significance are shown. The correlation has been performed using ClimateReanalyzer.Org, University of Maine, USA. Coloured dots indicate locations of WDC (red), Siple Dome (green), RICE (blue), TALDICE/Talos Dome (purple), and Taylor Dome (orange).**

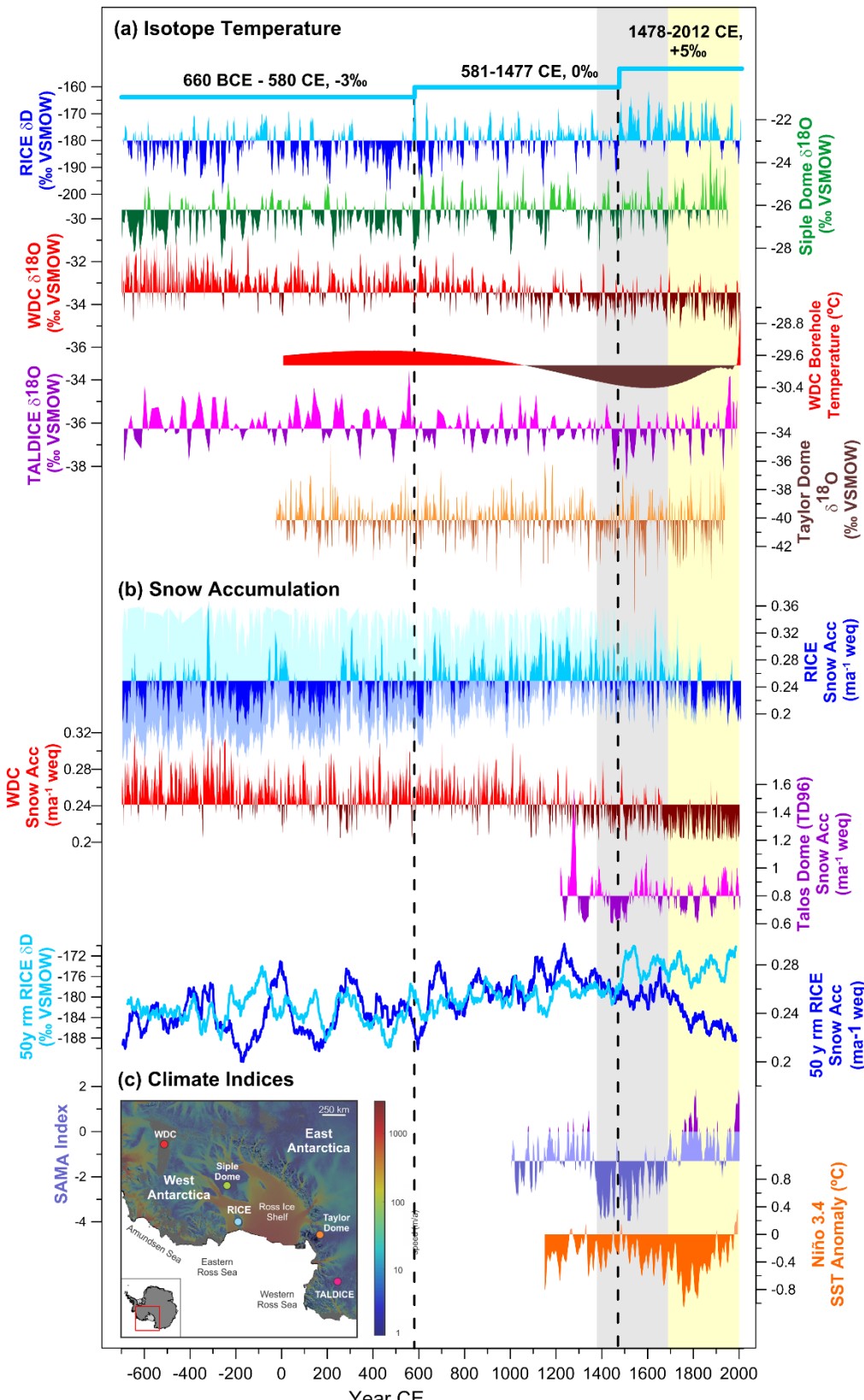

**Figure 6: a)** Isotope records for the past 2,700 years for RICE, Siple Dome (Brook et al., 2005), WDC (WAIS Divide Project Members, 2013), TALDICE (Stenni et al., 2011) and Taylor Dome (Steig et al. 1999, Steig et al. 2000, Sigl et al. 2014, PAGES2k Consortium 2017); **b)** snow accumulation data for RICE (Winstrup et al., 2017), WDC (Fudge et al., 2016), and Talos Dome (TD96). No snow accumulation data are available for Siple Dome or Taylor Dome; **c)** Reconstructions of Climate Indices for $SAM_A$ (Abram et al., 2014) and Niño 3.4 based on HadSST2 (Emile-Geay et al., 2013). Colour coding identifies above and below average values. Grey shaded area emphasises period of negative $SAM_A$, yellow shaded area emphasises period of synchronous warming at RICE, Siple Dome, WDC and TALDICE.

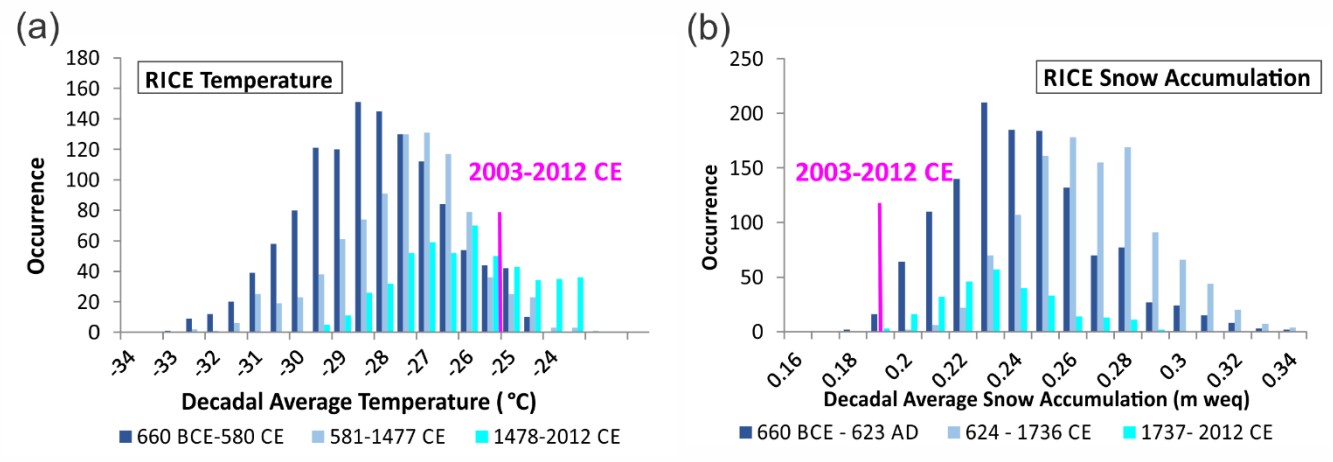

**Figure 7: a) Frequency occurrences of decadal temperature variations (10 year moving averages) as reconstructed from the RICE ice core are shown for three periods: 660 BCE to 580 CE – dark blue, 581-1477 CE – light blue, and 1478-2012 CE -cyan. The 10 year average temperature for the most recent decade contained in the record, 2002-2012, -25.78 deg C, is shown in pink. b) Frequency occurrences of decadal RICE snow accumulation variations (10 year moving averages for 660 CE to 623 CE – dark blue, 624-1736 CE – light blue, and 1737-2012 CE - cyan. The 10 year average for the most recent contained in the record, 2002-2012, 0.2m weq is shown in pink.**

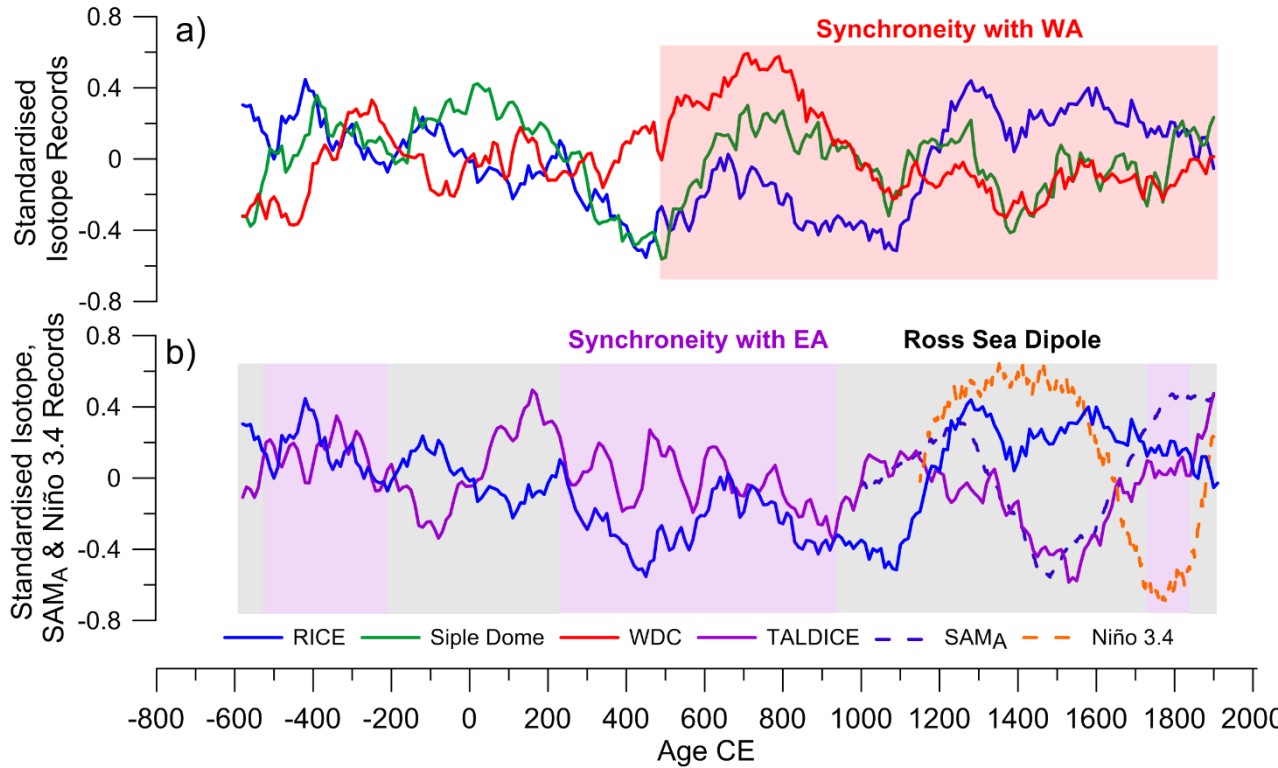

**Figure 8: Phasing of multi-decadal and centennial climate temperature variability at RICE, Siple Dome, WDC and TALDICE using detrended, normalised isotope records, smoothed with a 200-year moving average. RICE is compared with a) Siple Dome and WDC and b) TALDICE to investigate phase relationships of climate variability in the eastern Ross Sea (RICE, Siple Dome) with West (WDC) and East Antarctica (TALDICE). WA= West Antarctica, EA= East Antarctica. The red shading indicates periods of synchroneity of RICE and Siple Dome records with WA. Grey shading indicates time periods where RICE (eastern Ross Sea) shows an antiphase relationship (a Ross Sea Dipole) with TALDICE (western Ross Sea). Purple shading identifies times when RICE and EA are in phase. The normalised SAM$_A$ and Niño 3.4 records, smoothed with a 200-year moving average, are shown for comparison.**

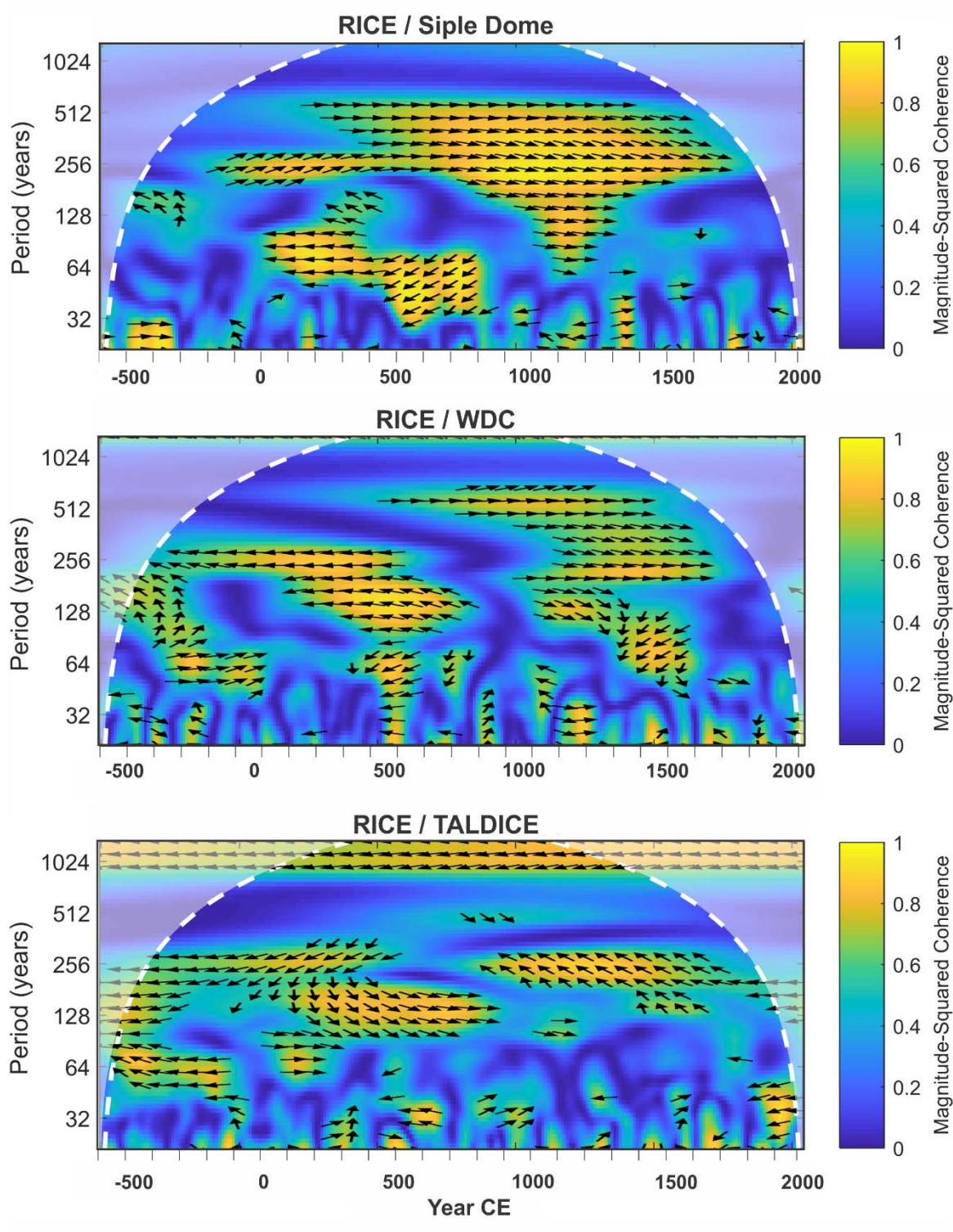

**Figure 9: Wavelet coherence and cross spectrum analysis of a) RICE δD and Siple Dome δ¹⁸O, b) RICE δD and WDC δ¹⁸O, c) RICE δD and TALDICE δ¹⁸O. The Analysis was conducted on decadally averaged, detrended data, smoothed with a 200 year moving average. The coherence is computed using the Morlet wavelet and is expressed as magnitude-squared coherence (msc). The phase of the wavelet cross-spectrum is provide for values over 0.6 msc using a Welch's overlapped averaged periodogram method (Kay, 1988;Rabiner et al., 1978). Arrows to the right indicate RICE is leading, arrows the left indicate RICE is lagging. An upright or downward arrow represents ¼ cycle difference.**