# Peer review of "The Ross Sea Dipole - Temperature, Snow Accumulation and Sea Ice Variability in the Ross Sea Region, Antarctica, over the Past 2,700 Years"

_Climate of the Past, 2017_

## Referee Comment (RC1) · Anonymous Referee #1 · 24 Aug 2017

Overall: This paper provides an analysis of a new dataset stemming from the Roosevelt Island Climate Evolution ice core. The authors identify how well this ice core compares with ERA-Int data, approximate the regional temperature and precipitation variability, and how well it compares with other proxy data in West Antarctica and portions of the western Ross Ice Shelf. The authors note an finding of a 'Ross Sea dipole' where there are periods of opposite relationships between the eastern and the western Ross Ice Shelf.

I think the paper holds promise, and certainly this new dataset needs to be presented and discussed widely. However, I find concern in interpreting the Ross Sea dipole to

the SAM.

The authors base the SAM connection with the Ross Sea dipole from one paper, Marshall and Thompson (2016), presumably Fig. 2b in this paper. This figure from Marshall and Thompson (2016) indeed shows opposite patterns of heat flux across the eastern and western Ross Ice shelf associated with the SAM. However, I'm not sure that this can easily be implied as consistent with the results from the anti-phase relationships observed in the proxy data presented in the paper, for a few reasons:

a) The relationships in the Marshall and Thompson (2016) paper were based on daily data, and the authors note that the heat flux relationships with the SAM are much weaker when integrated over time periods more than a week. It is therefore really hard to know if they still exist on annual mean data (let alone data that are smoothed with 200-year moving averages!). This dipole pattern with the SAM and the heat flux is also found in reanalysis data since 1979, which certainly can't tell us much about its persistence on timescales back more than 100 years.

b) Even if there were a dipole pattern associated with the SAM that persisted, there clearly isn't a dipole pattern with temperature and the SAM (Fig. 3b of Marshall and Thompson (2016) and many other works, including Marshall (2007), Thompson and Solomon (2002) etc.). In terms of temperature, the SAM exerts the same-sign relationship across the entire Ross Ice shelf, and West Antarctica and East Antarctica. For precipitation / accumulation, there may be more of a dipole like structure (this is nearly impossible to verify with observations or reanalyses), but so many local factors influence precipitation / accumulation that it is hard to say how robust any Ross Sea dipole pattern is.

c) Climate connections with the ASL, whether from ENSO or the SAM, show dipole patterns on much larger scales, with differences (in temperature, precipitation, winds, etc) occurring between the Antarctic Peninsula / eastern West Antarctica and the Ross Sea (Ice Shelf) / western West Antarctica. I don't know fully how a dipole pattern across

just the Ross Ice Shelf, from annual mean data or longer, could be related to these much larger-scale climate patterns, at least based on observations and contemporary reanalyses.

I therefore found the climate connections and their interpretation with the RICE data to be far too simplistic and an incorrect interpretation of one figure from Marshall and Thompson (2016). The authors need to revise this portion of the paper and better justify / support the pattern in relationship to the SAM, or simply not make claims that it is consistent with the SAM.

Minor comments:

Abstract, line 31: change 'Annual' to 'Annular'

Line pg 3: 26: gradient of what, exactly? Just pressure / height, or other fields?

Lines pg 3. 30-33: you should specify this is increase in total Antarctic SIE, as there are regional differences.

Figure 2: the color scale for the correlations is odd. It makes it challenging to see what the magnitude of the correlations are in the top panel. Even if they are significantly different from zero, a small correlation explains very little of the interannual variability and therefore may not be an ideal representation of temperature variability at other regions in West Antarctica or off the Ross Ice Shelf.

Fig 2e, discussion of temp. trends on pg. 6 lines 37-40 and pg. 7 lines 1-7: It is fair to say ERA-Int may not capture the correct trend at the RICE site, but why not compare with observations directly at McMurdo / Scott Base or any of the longer Wisconsin AWS records (Gill , Ferrell, etc.) on the Ross Ice Shelf? These are strongly correlated with the RICE site based on Fig. 2a. I think comparing with NB2014 is helpful, but I think it is a huge oversight to not do any comparisons with direct observations (you could even use the Byrd temperature record here).

Fig 3, discussion page 7 lines 30-31: ERA-Int could also be different in that it uses a

different snow density and/or conversion from precipitation to water equivalent. (something with the microphysics in ERA-Int model).

Figure 4: Also not particularly happy about the color scale here for the correlations.

Page 7, lines 37-38: It would be more instructive to say that the negative correlation includes regions of the South Pacific, Antarctic Peninsula, and eastern West Antarctica, rather than the 'ASL region' since the ASL varies its location from month to month, and the correlation is not significant across the entire region that the ASL may reside.

Page 9, lines 3-5: The Nino 3.4 and Nino 4 are close (and overlap partially), and are therefore strongly temporally correlated. However PSA1 and PSA2, by design through EOF, are uncorrelated in time and space. I don't think using Nino 4 for PSA2 is a good idea because of this.

Page 9, Table 2 ENSO correlations lines 10-15: In addition to differences in the phasing of ENSO and SAM, using annual means for ENSO is also compromising the correlations, since ENSO events wrap around a calendar year (peaking in December often). They are likely stronger on seasonal means; this should be mentioned.

---

## Short Comment (SC2) · 16 Sep 2017

Sebastian Luening

luening@uni-bremen.de

The new RICE ice core data are much welcome and add an important new Antarctic dataset. Looking at the past few millennia, the existing ice core isotope curves from Antarctica often differ greatly from each other, making it complicated to identify common trends. The Ross Sea Dipole described in the current manuscript highlights an important regional climate relationship which needs to be taken into account when extrapolating Antarctic palaeoclimate data across the continent. While some ice cores and sedimentary cores have larger-scale significance, other palaeoclimate records may only be of local character, which needs to be thoroughly worked out before at-

tempting full scale continent-wide palaeoclimate reconstructions.

The manuscript describes a new deuterium dataset. Oxygen isotope data are unfortunately not presented. This may be planned for a future paper. Nevertheless, a comparison of dD and d18O may be useful, because it appears that the two temperature proxy types may not always show the same evolution in the Ross Sea Region. The Taylor Dome Ice Core lies on the opposite, western side of the Ross Ice Shelf. The dD and d18O data have been archived here: http://isolab.ess.washington.edu/isolab/taylor/data/iso.html

I have plotted the dD and d18O curves for the past 1500 years (correlation attached, MCA=Medieval Climate Anomaly) and found some interesting opposing trends. Between 600-900 AD the two Taylor Dome temperature proxies show inverse behaviour. The same happens between 1000-1450 AD. Which of the two proxies shows the real temperature? It would be important to better understand these opposing trends in the Taylor Dome Ice Core. Does a similar phenomenon occur in the RICE ice core? Notably, dD and d18O curves are very similar in the Siple Dome.

Also plotted in the attached correlation is the Victoria Lower Glacier, VLG (Bertler et al. 2011) which shows some peculiar similariaties and differences with the other ice cores mentioned in the current manuscript. It may be worth comparing the VLG with the other ice cores.

[Figure]

Fig. 1.

---

## Referee Comment (RC2) · Anonymous Referee #2 · 14 Oct 2017

In this study, the authors are presenting high-resolution isotopic (dD) and snow accumulation rate records of a new ice core (RICE) drilled on the Roosevelt Island in the eastern Ross Sea and covering in its upper part the last 2700 years. The authors are comparing these new records to other ice core records present in the near-by areas interpreting these records in terms of the climate variability which in these areas (Eastern and Western Ross Sea) is characterized by a climate pattern referred to as the Ross Sea dipole. The paper is noteworthy and the authors are doing a good job in calibrating the new records against the Era-interim re-analysis data (temperature and precipitation) as well as to other climate indexes as SAM, SOI, Nino3.4, IPO, as well as Sea Ice extent. However, some methodological questions are arising in this part (see

below). The manuscript is quite well structured and the topic is appropriate for Climate of the Past. Nevertheless, the authors should consider the comments reported below before resubmitting a revised version. One general comments refers to the fact that the manuscript (Winstrup et al., CPD) presenting the ice core dating, on which most of the interpreted data are relying on, is not published yet. There are also other papers (one is Emanuelsson et al.), to which the authors are referring that are still at the submission7review stage, please check and update. General as well as detailed comments are reported below. Page 3, line 3: may you check this sentence? Over the observational period (satellite era) the sea ice should be increasing in the Ross Sea sector and decreasing in the Amundsen–Bellingshausen sector. Page 3, line 11: here the authors are saying that they will compare these new records to other ones existing in the region but it is not clear why they do not consider the Taylor Dome ice core record. Page 6, lines 27-32. Here the authors are optimizing the dD/T relation to the age scale. I am wondering why the authors did not consider optimizing the snow accumulation rate to the ERA-I precipitation rather than the dD/T relation. In fact, it is known and also the authors are clearly showing this, that the isotopic composition rely not only on the temperature but also on other circulation-related factors. How, this choice is affecting the climate interpretation? The authors should answer to this comment. Page 6, line 31: Figure 2b should be 2c. Page 6, line 35: in the Masson-Delmotte paper only spatial d/T slope are considered. Page 7, line 5: Is the lack of correlation valid also considering only the 1979-2012? Page 7, line 6: RICE dD should be RICE dDo. Page 7, lines 11-14: Is this strong accumulation rate gradient suggesting possible movements of the dome in the past. May you explain this? Page 7, line 37-38: the region which exhibits a negative correlation seems to be at lower latitudes than the ASL, at least looking at the figure …. Page 8, lines 10-12: the strong impact of blocking events at this site would support my comment above (Page 6, lines 27-32). Page 8, line 13: the negative correlation seems to interest more the Amundsen Sea. Page 8, line 17: are polynyas resolved by the model data? Page 8, line 21: I would move this sentence: "We focus …." at the beginning of the paragraph. Page 8, line 20: please add

a URL link to these data (SIEj). Page 9, lines 1-7: please add data citation or URL in Data Availability section for all the climate indexes used. Page 9, line 9: why using the SAMa index instead of SAM for this period? Are these two indexes the same over this period? Page 9, line 13: ….... (but not with Rice snow accumulation)... See again my comment above (Page 6, lines 27-32). Page 9, lines 22-24: again why not using Taylor Dome? On the other side, regarding the use of TALDICE data: I would suggest to use Talos Dome (89 m core) isotopic and snow accumulation rate data (Stenni et al., 2002) rather than TALDICE for this recent period. The TALDICE data are low res-olution data and the snow accumulation rate, here considered, comes from the dating model and as such implicitly connected to the isotopic record from which it is derived. On the contrary, the isotopic and snow accumulation rate records from the Talos Dome core, although limited to the past 800 years, are high-resolution data and the dating has been performed by nssSO4 annual data constrained by the volcanic chronology. Moreover, the TALDICE data, here reported are on the Buiron et al. (2011) age scale which has been replaced by the AICC-2012 chronology. Between the two there are in some cases differences up to 150 years than for the purposes of this paper could be important. So, the authors could consider using both isotopic data sets (Talos Dome and TALDICE but the latter on the AICC-2012 age scale) and the snow accumulation rate from only Talos Dome. An alternative to the AICC2012 age scale for TALDICE is the Severi et al. (2012, CP) chronology, which uses the volcanic synchronisation between the TALDICE and the EPICA Dome C ice cores and in practice identical to AICC2012 age scale. Page 10, line 14: the onset of a decline in WDC isotopes at 579 CE is not clear from the figure ... Page 10, lines 25-26: the sentence "From the 17th century ...... " it is not very clear, at least looking at the figure, and the word in phase seems to be not the correct word to use.... Page 10, line 30: TALDICE data: see the comments above. Page 10, lines 37: from here ("until the 15th century …..") to the end of the paragraph it is not easy to follow the reasoning... Page 11, line 19: how this date (1367 +/-12 CE) is chosen? Not clear. Please, explain …..I suppose from the SAMA record... Page 12, line 24: increased marine air mass intrusions: from borehole
or isotopic record? Page 13, line 30: ….... are anti-correlated ….... before the authors were claiming that they are correlated (see fig 8) at least from 400 to 1900 CE. Page 17, line 1: the author list of this reference (Jones et al) seems to be not complete. Page 22 Table 1: I would suggest adding the resolution of the different records. Again, for TALDICE I would suggest using the AICC2012 or the Severi et al. (2012) age scales. Always in the Table 1: the reference WAIS Divide Members, 2013 refers to WDC and not Siple Dome. Page 26, Figure 4 caption: add the explanation for panels c and d. Page 28, figure 6 caption: for RICE, are these decadal averages or what? Page 30, figure 8 caption: please add that these are isotopic data.

––––––––––––––––––––––––––––

---

## Author Comment (AC1) · 10 Nov 2017

**Authors' Response:**

We thank the Reviewer for the comments and suggestions. In the text below, we outline our response in blue.

**Anonymous Referee #1**

Overall: This paper provides an analysis of a new dataset stemming from the Roosevelt Island Climate Evolution ice core. The authors identify how well this ice core compares with ERA-Int data, approximate the regional temperature and precipitation variability, and how well it compares with other proxy data in West Antarctica and portions of the western Ross Ice Shelf. The authors note an finding of a 'Ross Sea dipole' where there are periods of opposite relationships between the eastern and the western Ross Ice Shelf.
I think the paper holds promise, and certainly this new dataset needs to be presented and discussed widely. However, I find concern in interpreting the Ross Sea dipole to the SAM.

The authors base the SAM connection with the Ross Sea dipole from one paper, Marshall and Thompson (2016), presumably Fig. 2b in this paper. This figure from Marshall and Thompson (2016) indeed shows opposite patterns of heat flux across the eastern and western Ross Ice shelf associated with the SAM. However, I'm not sure that this can easily be implied as consistent with the results from the anti-phase relationships observed in the proxy data presented in the paper, for a few reasons:

a) The relationships in the Marshall and Thompson (2016) paper were based on daily data, and the authors note that the heat flux relationships with the SAM are much weaker when integrated over time periods more than a week. It is therefore really hard to know if they still exist on annual mean data (let alone data that are smoothed with 200-year moving averages!). This dipole pattern with the SAM and the heat flux is also found in reanalysis data since 1979, which certainly can't tell us much about its persistence on timescales back more than 100 years.
b) Even if there were a dipole pattern associated with the SAM that persisted, there clearly isn't a dipole pattern with temperature and the SAM (Fig. 3b of Marshall and Thompson (2016) and many other works, including Marshall (2007), Thompson and Solomon (2002) etc.). In terms of temperature, the SAM exerts the same-sign relationship across the entire Ross Ice shelf, and West Antarctica and East Antarctica. For precipitation / accumulation, there may be more of a dipole like structure (this is nearly impossible to verify with observations or reanalyses), but so many local factors influence precipitation / accumulation that it is hard to say how robust any Ross Sea dipole pattern is.
c) Climate connections with the ASL, whether from ENSO or the SAM, show dipole patterns on much larger scales, with differences (in temperature, precipitation, winds, etc) occurring between the Antarctic Peninsula / eastern West Antarctica and the Ross Sea (Ice Shelf) / western West Antarctica. I don't know fully how a dipole pattern across just the Ross Ice Shelf, from annual mean data or longer, could be related to these much larger-scale climate patterns, at least based on observations and contemporary reanalyses.

I therefore found the climate connections and their interpretation with the RICE data to be far too simplistic and an incorrect interpretation of one figure from Marshall and Thompson (2016). The authors need to revise this portion of the paper and better justify / support the pattern in relationship to the SAM, or simply not make claims that it is consistent with the SAM.

Our hypothesis that the Ross Sea Dipole might be, at least in part, an expression of the regional influence of the SAM, is based on a range of observations, with the principal evidence derived from the comparison of the 1,000 year-long SAM reconstruction ($SAM_A$) by Abram et al. 2014. During the most negative phase of the $SAM_A$, we observe cooler (warmer) conditions

in the western (eastern) Ross Sea (Figure 6). We further investigate this relationship by comparing detrended isotope records from RICE, Siple Dome, WDC and TALDICE, smoothed with a 200 year moving average (Revised Figure 8). Please note that in the revised Fig.8 we use an updated from Severi et al, 2012 age scale for TALDICE. The original and revised Figure 8 identified time periods dominated by opposing temperature trends in the eastern / western Ross Sea. Instead of time periods of known and hypothesised past negative and positive SAM time periods, we have modified the figure to show the $SAM_A$ reconstruction smoothed with a 200 year running mean. This replaces frames used in the original figure, which identified the the two periods of prolonged positive and negative SAM phasing as recorded in the $SAM_A$ record. The revised figure illustrates the relationship between $SAM_A$ and the phasing / occurrence of the Ross Sea Dipole. In the manuscript, we explore a possible mechanism to explain the influence of the SAM on the spatial temperature pattern influenced by the heat flux pattern observed by Marshall and Thompson (2016) and as noted by Reviewer 1, the influence of the SAM producing a dipole in the meridional heat flux across the Ross Ice Shelf can be observed in the ERAi data for the 1979-2012 time period. This suggests that the SAM - dipole heat flux relationship remained robust despite changes in the phasing of ENSO and the IPO. Marshall and Thompson (2016) find that this dipole is not apparent in temperature. We don't necessarily see this contradicting our hypothesis as on short time periods other drivers, in particular sea ice conditions have a strong, perhaps masking influence. But if changes in heat flux and meridional winds persist over longer time periods, we would expect those changes to lead to changes in sea ice extent, temperature and snow accumulation in the Ross Sea.

However, we agree with Reviewer 1 that we did not provide sufficient evidence to suggest a causality. For this reason, we have revised the manuscript to remove statements suggesting causality and instead only note the co-variance between the $SAM_A$ record and the reconstructed spatial temperature pattern.

[Figure]

**Revised Figure 1: Phasing of multi-decadal and centennial climate variability at RICE, Siple Dome, WDC and TALDICE using detrended, normalised isotope records smoothed with a 200-year moving average. RICE and Siple Dome are compared with a) WDC and b) TALDICE to investigate phase relationships of climate variability in the eastern Ross Sea with West (WDC) and East Antarctica (TALDICE). WA= West Antarctica, EA= East Antarctica. The detrended, normalised $SAM_A$ record (Abram et al. 2014, light blue) has been smoothed with a 200-year moving**

**average and is shown in panel (b). Shaded periods indicate synchroneity of RICE (and Siple Dome) data with WA (red box) or EA (purple box). Grey shading indicate time periods where RICE (eastern Ross Sea) shows an antiphase relationship (a Ross Sea Dipole) with TALDICE (western Ross Sea).**

Minor comments:

Abstract, line 31: change 'Annual' to 'Annular'

Done

Line pg 3: 26: gradient of what, exactly? Just pressure / height, or other fields?

Changed to 'pressure gradient'

Lines pg 3. 30-33: you should specify this is increase in total Antarctic SIE, as there are regional differences.

Pg 3, line 31 - Changed to 'the total Antarctic SIE increase'

Figure 2: the color scale for the correlations is odd. It makes it challenging to see what the magnitude of the correlations are in the top panel. Even if they are significantly different from zero, a small correlation explains very little of the interannual variability and therefore may not be an ideal representation of temperature variability at other regions in West Antarctica or off the Ross Ice Shelf.

The spatial correlation pattern in panel (a), (b), and (c) are shown to highlight and compare the spatial representativeness of records derived from the RICE site (extracted ERAi data) and actual RICE data ($\delta$D, snow accumulation). Only correlations significant at >95% are included. While weak correlations can be useful, correlations at r> +0.4 and r< -0.4 are especially distinct with the chosen colour scale changing from red to yellow and from turquoise to blue, respectively. We feel that this colour scheme therefore provides an accessible and clear representation of both the pattern and the strength of the correlation.

Fig 2e, discussion of temp. trends on pg. 6 lines 37-40 and pg. 7 lines 1-7: It is fair to say ERA-Int may not capture the correct trend at the RICE site, but why not compare with observations directly at McMurdo / Scott Base or any of the longer Wisconsin AWS records (Gill , Ferrell, etc.) on the Ross Ice Shelf? These are strongly correlated with the RICE site based on Fig. 2a. I think comparing with NB2014 is helpful, but I think it is a huge oversight to not do any comparisons with direct observations (you could even use the Byrd temperature record here).

There are two principal reasons why we did not use weather station data:
- Records in the vicinity are either short and/or suffer from large data gaps (Margaret, Gill, Ferrell AWS, Siple Station and the original Byrd Station)
- Station records (McMurdo, Scott Base) are at the opposing site of the observed Ross Sea Dipole and the reconstructed Byrd Station record also falls at the margin of the correlation pattern.

However, we agree that it would be useful to show the comparisons. We propose to include the Figure S1 and Table S1 in the supplementary information of the manuscript. For clarity, the comparison in Figure S1 is shown for the raw data covering 1957-2012 and for standardised data for 1957-2012 and 1979-2012. The comparison highlights that while all data sets agree on the occurrence of particularly extreme cold (i.e. 2004, 2010) or warm (i.e. 1980) years, there is large spatial, interannual variability across the data sets. Table S1 shows that for the satellite period (1979-2012), only the correlation between RICE $\delta$D and ERAi is

statistically significant. If the 1957-2012 time period is considered, the NB2014 reanalysis data set also becomes significant, but only at p<0.1, which is a level not considered in our original manuscript.

[Figure]

Figure S1: Comparison of temperature data from Antarctic Stations, remote Antarctic Weather Stations (AWS), and reanalysis products with δD RICE data. Origin of the data is referenced in Table R1-1. Panel (a) shows the actual data, panels (b) and (c) show the comparison of the standardised records for the time periods 1957-2012 and 1975-2012, respectively.

Table S1: Pearson correlation coefficient (r) and significance values (p) for correlations between RICE δD record and relevant observational records from automatic weather stations and reanalysis data. Data for the reconstructed Byrd Station meteorological data (Bromwich et al., 2013) are accessed via the Byrd Polar Research Centre, Polar Meteorological Group, Ohio State University (http://www.polarmet.osu.edu/datasets/Byrd_recon/). Weather station data for Ferrell, Gill, and Margaret AWS are accessed via Antarctic Meteorological Research Center and Automatic Weather Station Project (https://amrc.ssec.wisc.edu). Data for McMurdo Station and Scott Base are accessed via the MET-READER (https://legacy.bas.ac.uk/met/READER/data.html). The number of years of observations represents the total number of years which contain monthly averages for each month of a calendar year. Only years with 12 monthly values are included in the correlation.

| Correlation with RICE δD | Location (lat/long) | Elevation (m asl) | Time Period Overlap | No of Years of Observations | r | p-Value |
|---|---|---|---|---|---|---|
| Byrd Station (revised by Bromwich et al. | 80.0° S 120.0° W | 1515 | 1957-2012 | 56 | 0.05 | 0.72 |
| Ferrell AWS | 77.9° S 170.8° E | 45 | 1982-2012 | 17 | 0.30 | 0.24 |
| Gill AWS | 79.9° S 178.6° W | 53 | 1987-2012 | 20 | 0.24 | 0.31 |
| Margaret AWS | 80.° S 165.0° W | 67 | 2009-2012 | 4 | 0.33 | 0.67 |
| McMurdo AWS | 77.9° S 166.7° E | 24 | 1957-2012 | 48 | 0.11 | 0.46 |
| Scott Base AWS | 77.9° S 166.7° E | 16 | 1958-2009 | 51 | 0.02 | 0.90 |
| Siple Station AWS | 75.9° S 84.0° W | 1054 | 1982-1992 | 5 | 0.02 | 0.97 |
| NB2014 | Extracted for nearest grid point to RICE location 79.39° S / 161.71° W | 550 | 1979-2012 | 34 | 0.26 | 0.14 |
| NB2014 | Extracted for nearest grid point to RICE location 79.39° S / 161.71° W | | 1958-2012 | 55 | **0.23** | **0.09** |
| ERAi | Extracted for nearest grid point to RICE location 79.39° S / 161.71° W | | 1979-2012 | 34 | **0.42** | **0.01** |

Fig 3, discussion page 7 lines 30-31: ERA-Int could also be different in that it uses a different snow density and/or conversion from precipitation to water equivalent. (something with the microphysics in ERA-Int model).

Agreed. We have added the sentence in line 31: '…and the actual drill site location, as well as differences in assumed snow densities, or different methodologies in the conversion from precipitation to water equivalent units.'

Figure 4: Also not particularly happy about the color scale here for the correlations.

We make the same argument as for Figure 2 above.

Page 7, lines 37-38: It would be more instructive to say that the negative correlation includes regions of the South Pacific, Antarctic Peninsula, and eastern West Antarctica, rather than the

'ASL region' since the ASL varies its location from month to month, and the correlation is not significant across the entire region that the ASL may reside.

Agreed. We have changed the wording accordingly: "A negative correlation is found in the regions of the South Pacific, Antarctic Peninsula and eastern West Antarctica"

Page 9, lines 3-5: The Nino 3.4 and Nino 4 are close (and overlap partially), and are therefore strongly temporally correlated. However PSA1 and PSA2, by design through EOF, are uncorrelated in time and space. I don't think using Nino 4 for PSA2 is a good idea because of this.

Agreed. We have revised the text in the manuscript to refer to the El Niño regions 3.4 and 4 and the resulting Rossby wave propagations.

Page 9, Table 2 ENSO correlations lines 10-15: In addition to differences in the phasing of ENSO and SAM, using annual means for ENSO is also compromising the correlations, since ENSO events wrap around a calendar year (peaking in December often). They are likely stronger on seasonal means; this should be mentioned.

As outlined in our manuscript, we are cautious to use RICE     seasonal means because of the variable frequency and intensity of precipitation events which have the potential to lead to seasonal biases. We have added the following sentence on page 9, line 14: 'Correlations using seasonal instead of annual averages might be more suitable to identify a linear relationship between RICE records and ENSO events, which usually peak during the austral summer, in particular December (Turner et al. 2004). However, we refrain from using seasonal means because of the variable frequency and intensity of precipitation events at Roosevelt Island which have the potential to lead to seasonal biases, as outlined in section 4.2."

We will add the following reference: Turner, J.: The El Niño–Southern Oscillation and Antarctica, International Journal of Climatology, 24, 1-31, 10.1002/joc.965, 2004.

---

## Author Comment (AC2) · 10 Nov 2017

**Authors' Response:**

We thank the Reviewer for the comments and suggestions. In the text below, we outline our response in blue.

**Anonymous Referee #2**

In this study, the authors are presenting high-resolution isotopic (dD) and snow accumulation rate records of a new ice core (RICE) drilled on the Roosevelt Island in the eastern Ross Sea and covering in its upper part the last 2700 years. The authors are comparing these new records to other ice core records present in the near-by areas interpreting these records in terms of the climate variability which in these areas (Eastern and Western Ross Sea) is characterized by a climate pattern referred to as the Ross Sea dipole. The paper is noteworthy and the authors are doing a good job in calibrating the new records against the Era-interim re-analysis data (temperature and precipitation) as well as to other climate indexes as SAM, SOI, Nino3.4, IPO, as well as Sea Ice extent. However, some methodological questions are arising in this part (see C1 below). The manuscript is quite well structured and the topic is appropriate for Climate of the Past. Nevertheless, the authors should consider the comments reported below before resubmitting a revised version.

One general comments refers to the fact that the manuscript (Winstrup et al., CPD) presenting the ice core dating, on which most of the interpreted data are relying on, is not published yet.

This publication is now in review and can be accessed here: https://www.clim-past-discuss.net/cp-2017-101/ including comments from two reviewers. We will update the reference to that publication accordingly.

There are also other papers (one is Emanuelsson et al.), to which the authors are referring that are still at the submission7review stage, please check and update.

Three additional references used in our manuscript are still in review: Emanuelsson et al. (in review), Keller et al. (in review), and Pyne et al. (in review). We will update these references as they become available. In the meantime, we offer to provide copies of the manuscripts to the Reviewer.

General as well as detailed comments are reported below.

Page 3, line 3: may you check this sentence? Over the observational period (satellite era) the sea ice should be increasing in the Ross Sea sector and decreasing in the Amundsen–Bellingshausen sector.

We have corrected the sentence to state "...changes in sea ice (wind driven, regional decreases and increases in the Amundsen and Ross Seas, respectively."

Page 3, line 11: here the authors are saying that they will compare these new records to other ones existing in the region but it is not clear why they do not consider the Taylor Dome ice core record.

The Taylor Dome age scale is constrained for the past 3,000 years with a flow model, which assumes a constant snow accumulation rate and lacks independent age benchmarks. The record prior to 3,000 years ago has an improved age scale due to ties with Greenland records through the correlation of their respective methane data. The lack of age control points in the past 3,000 year record make the Taylor Dome data unsuitable to investigate the timing and phasing of climatic shifts on decadal or centennial time scales. However, we agree with Reviewer 2 that it is useful to show the Taylor Dome

isotope data to support longer term temperature trends in the region and for this reason we now include the record in Figure 6. Like the Talos Dome data, the Taylor Dome isotope record shows no trend over the past 2,700 years. We have revised the manuscript to include the description of the Taylor Dome isotope record.

Page 6, lines 27-32. Here the authors are optimizing the dD/T relation to the age scale. I am wondering why the authors did not consider optimizing the snow accumulation rate to the ERA-I precipitation rather than the dD/T relation. In fact, it is known and also the authors are clearly showing this, that the isotopic composition rely not only on the temperature but also on other circulation-related factors. How, this choice is affecting the climate interpretation? The authors should answer to this comment.

The brevity of the overlap between the RICE data and reanalysis products (1979-2012) provides some challenges for the assessment of correlations between the proxy records and climate parameters. Using an optimisation approach, we investigated whether the shift of up to  $\pm 1.3$  years (within the age scale uncertainty of  $\pm 2$  years) can significantly impact the results. The optimisation increased the correlation from r=0.42 to r=0.66) between temperature and isotope data. In a second step, we assessed how this optimised age model solution affects other parameters. The correlation between the snow accumulation data and the ERAi precipitation data on the RICE17 age scale is already high with r=0.60. Using the same approach marginally improves the correlation value between snow accumulation and ERAi precipitation (r=0.62). However, here we wanted to test whether the age scale optimised for the isotope data would positively or negatively influence the correlations for other parameters. Using the isotope optimised age model, ERAi precipitation and RICE Acc0 remains statistically significant but is reduced to r=0.42. We conclude from this that the optimised age solution is not superior to the original RICE17 age scale. We also conclude that the RICE data are useful parameters to reconstruct both temperature and snow accumulation within the given uncertainties and that the obtained correlations values are likely limited by the brevity of the records.

Page 6, line 31: Figure 2b should be 2c.

Done.

Page 6, line 35: in the Masson-Delmotte paper only spatial d/T slope are considered.

We have broadened the cited  $\delta/T$  slopes to include the range reported by Schneider et al. 2005 (interannual  $\delta/T$  slopes) which includes the range from 2.9 to 3.4 ‰ per degC.

**Page 7, line 5: Is the lack of correlation valid also considering only the 1979-2012?**

We agree with Reviewer 2 that the brevity of the records limits the usefulness of the correlations. If the longer time frame from 1958-2012 is considered, the correlation between RICE  $\delta D$  and NB2014 becomes weakly statistically significant (r=0.23, p=0.09). However, it is curious that the ERAi and NB2014, which both use similar inputs of observational and satellite data, do not correlate at the RICE site. The Roosevelt Island topography currently cannot be adequately represented in the data products because of their grid resolution. Yet, the topography might have sufficient vertical profile to influence the inversion layer thickness and precipitation pattern that alter the local conditions from the precipitation pattern of the surrounding Ross Ice Shelf. However, the comparison between ERAi and NB2014 is beyond the scope of this paper. For this reason, we do not offer a hypothesis for the cause of the observed difference. Furthermore, we note that statistically significant relationships between RICE  $\delta$ D and ENSO indices (SOI, Niño 3.4 and 4) are observed when a shorter time frame 1979-2009 period is considered. In 2009, the IPO changes sign which influences the teleconnection between the ENSO signal and the South Pacific climatic response, perhaps masking this important ENSO-RICE relationship. We have added a sentence describing this observation. However, the observational time series are too short to quantify the impact of the changing IPO with RICE  $\delta$ D data and is beyond the scope of this manuscript.

Page 7, line 6: RICE dD should be RICE dDo.

We changed it to "RICE dD and dDo" (as neither correlates)

Page 7, lines 11-14: Is this strong accumulation rate gradient suggesting possible movements of the dome in the past. May you explain this?

It is possible that the divide has migrated in the past as a result of an imbalance in the ice flux on either side of the divide, but this is not necessarily caused by changes in the accumulation gradient; it could equally well arise from changes in the efflux across the grounding-line, caused by changes in buttressing. The small migration in divide position suggests that neither accumulation gradient nor grounding line efflux have changed very much; this implies that the buttressing has not changed significantly either.

Page 7, line 37-38: the region which exhibits a negative correlation seems to be at lower latitudes than the ASL, at least looking at the figure . . ..

Agreed. We changed the description to: "A negative correlation is found in the regions of the South Pacific, Antarctic Peninsula and the eastern West Antarctica".

Page 8, lines 10-12: the strong impact of blocking events at this site would support my comment above (Page 6, lines 27-32).

As described above, ERAi precipitation and RICE snow accumulation correlate at r=0.60. Using the same optimisation methodology as tested for the RICE  $\delta D$  record only marginally improves the correlation (r=0.62). For this reason, no optimisation was carried out.

Page 8, line 13: the negative correlation seems to interest more the Amundsen Sea. Page 8, line 17: are polynyas resolved by the model data?

We have clarified the paragraph: "Snow accumulation at RICE is negatively correlated with SIE in the Ross Sea and northern Amundsen Sea region (Figure 5a), which predominantly represents sea ice exported from the Ross Sea. We observe that years of increased (decreased) SIE leading to reduced (increased) accumulation at RICE, confirming the sensitivity of moisture-bearing marine air mass intrusions to local ocean moisture sources and hence regional SIE. The correlation between ERAi SIE and the optimised RICE  $\delta D_0$  record (Figure 5b) similarly shows a negative correlation of SIE in the Ross and northern Amundsen Sea (perhaps with the exception of the Ross and Terra Nova polynyas) suggesting more depleted (enriched) values during years of increased (reduced) SIE." We do not use a model in our analysis but the large and generally persistent Ross Sea and Terra Nova polynyas are well observed in satellite data.

Page 8, line 21: I would move this sentence: "We focus . . .." at the beginning of the paragraph.

Agreed. We have revised the sentence to: "We focus on the SIE in the Ross and Amundsen Seas to investigate the relationship between the SIE and the temperature and precipitation at Roosevelt Island. In Figure 5 c and d, the Ross-Amundsen Sea SIE index (SIEJ), developed by Jones et al. (2016), is correlated with ERAi SAT and precipitation data."

Page 8, line 20: please add C2 a URL link to these data (SIEj).

We will add the sources of all data and indices used in this manuscript under the section "Data Availability" once the paper has been accepted for publication.

Page 9, lines 1-7: please add data citation or URL in Data Availability section for all the climate indexes used.

Please see comment above.

Page 9, line 9: why using the SAMa index instead of SAM for this period? Are these two indexes the same over this period?

We focus here on the Abram et al. (2014) SAM index (SAMA) as it provides a reconstruction of the SAM to 1000 CE. We use the SAMA index for the 1979-2012 correlations to assess the fidelity of the relationship between this index and our records to support a comparison over past centuries. The Marshall SAM record, which is based on meteorological observations, starts in 1957 and thus cannot support comparisons beyond the modern frame. During the common time period, the Marshall and Abram SAM indices are similar but also revealed some interesting differences as shown in the graph below. However, a detailed discussion of those difference is beyond the scope of this paper.

Figure R2-1: Comparison between the Marshall and Abram SAM indices (r=0.75).

Page 9, line 13: . . ... (but not with Rice snow accumulation). . . See again my comment above (Page 6, lines 27-32).

We do not use a snow accumulation record optimised to correlate with ERAi snow precipitation because the optimisation only led to a marginal improvement of the already high correlation value of r=0.60). Furthermore, the optimisation only improved either the correlation with temperature or the correlation with snow accumulation but not both with the same optimisation.

Page 9, lines 22-24: again why not using Taylor Dome? On the other side, regarding the use of TALDICE data: I would suggest to use Talos Dome (89 m core) isotopic and snow accumulation rate data (Stenni et al., 2002) rather than TALDICE for this recent period. The TALDICE data are low resolution data and the snow accumulation rate, here considered, comes from the dating model and as such implicitly connected to the isotopic record from which it is derived. On the contrary, the isotopic and snow accumulation rate records from the Talos Dome core, although limited to the past 800 years, are high-resolution data and the dating has been performed by nssSO4 annual data constrained by the volcanic chronology. Moreover, the TALDICE data, here reported are on the Buiron et al. (2011) age scale which has been replaced by the AICC-2012 chronology. Between the two there are in some cases differences up to 150 years than for the purposes of this paper could be important. So, the authors could consider using both isotopic data sets (Talos Dome and TALDICE but the latter on the AICC-2012 age scale) and the snow accumulation rate from only Talos Dome. An alternative to the AICC2012 age scale for TALDICE is the Severi et al. (2012, CP) chronology, which uses the volcanic synchronisation between the TALDICE and the EPICA Dome C ice cores and in practice identical to AICC2012 age scale.

We have replaced the TALDICE (a) snow accumulation and (b)  $\delta^{18}$ O records on the Buiron et al. 2011 age scale with (a) the Talos Dome (TD96) record for snow accumulation on the Severi et al. 2012 age scale and (b) TALDICE  $\delta^{18}$ O on the Severi et al. 2012 age scale for the temperature reconstruction. We have updated the relevant figures (Figure 6, 8, and 9). The use of the updated age scale did not change the conclusions of our interpretation but led to a clearer dipole pattern shown in revised Figure 8. The Talos Dome (TD96) snow accumulation record spans a shorter time period that the TALDICE data. We have revised the text to reflect this. Please see our comment below.

---

## Author Comment (AC3) · 10 Nov 2017

Hi Sebastian, Thank you for your comment and suggestions. With regards to your suggestion to compare the RICE data with the Victoria Lower Glacier (VLG) and Taylor Dome records, please note that we include in our manuscript the comparison between the RICE and VLG records (along other records) – please see page 11, lines 30-38 and also page 10, lines 7-9. The comparison discusses the spatial differences, including the MWP/LIA temperature anomalies. So your point is well taken but we feel it is already addressed in our manuscript. With regards to the Taylor Dome ice core, we initially did not include this record here because of its somewhat larger age uncertainty.

However, prompted by your comment and by Reviewer 2, we have now added the the Taylor Dome record in Figure 6 for comparison of the long term trends, which is very useful. With regards to the opposing trends between Taylor Dome d18O and dD values – we suspect that there might be an issue with the data in this file or the age scale used to plot the Taylor Dome records. It is surprising that d18O and dD values are so different. We would suggest that you contact the owners of the data to check whether perhaps the correct/same age scale is used for both files. It is inconceivable that dD and d18O measured on the same sample and plotted on the same age scale would show a distinct phase relationship. Below we plotted the d18O and dD data for VLG and RICE for comparison. As you can see from the figure, the two data sets (dD and d18O of each core) look almost indistinguishable. Of course when you calculate the deuterium excess, it becomes apparent that they are not identical. But we expect dD and d18O to be highly correlated (i.e. r>0.95).

———————————————————

[Figure]

**Fig. 1.** Figure SC_1: Comparison between dD and d18O for RICE and VLG. Colour coding – red (blue) values indicate values above (below) the long term average. Both records are plotted on the CE time scale.

---

## Author Comment (AC5) · 13 Nov 2017

**Corrections to the Manuscript**

We erroneously used in the comparison between the reanalysis and RICE data which stemmed from the annually resampled RICE  $\delta D$  record instead of the annually averaged RICE  $\delta D$  record. However, correctly using the annually averaged RICE  $\delta D$  record does not change our interpretation or conclusions.

Below we outline the revised correlations for Table 2, which include a) the annually averaged RICE  $\delta D$  record, b) the optimised RICE  $\delta D_0$  record and c) the optimised RICE snow accumulation (RICE Acc0) record. The RICE Acc0 record is calculated using the age scale used to find the optimal correlation between ERAi SAT and RICE  $\delta D$  data. The comparison with the correct data shows that in most cases the correlations either did not change or improved, with exception of the correlation between RICE  $\delta D/\delta D_0$  and SAT/Precipitation where correlations weakened. Original values that required recalculation are shown in 'blue' while the new, revised values are shown in 'red'

We also show the revised spatial correlation fields and time series comparison for Figure 2 (RICE  $\delta D/\delta D_0$  with ERAi SAT) and Figure 4 (RICE Acc/Acc0 with ERAi Precipitation)

**Table 1:** Overview of correlation coefficients for annual means of the common time period 1979-2012 between climate parameters, proxies and indices: the original RICE ( $\delta$ D) and optimised ( $\delta$ Do) data (this paper), the original RICE snow accumulation data (RICE Acc, Winstrup et al., in review) and data adjusted to the revised age scale of  $\delta$ Do – Acco, ERAi Surface Temperature (ERAi SAT) and Precipitation (ERAi Precip), Dee at el., 2011), Ross/Amundsen Sea Sea Ice Extent (SIEJ, Jones et al., 2016), Southern Annular Mode Index (SAMA, Abram et al., 2014), Southern Oscillation Index (SOI, Trenberth and Stepaniak, 2001), Niño 4 Index (Trenberth and Stepaniak, 2001) and Niño 3.4 (Emile-Geay et al., 2013), Interdecadal Pacific Oscillation Index (IPO, Henley et al., 2015), and the near-surface Antarctic temperature reconstruction (NB2014, Nicolas and Bromwich, 2014). Significance values are adjusted for degree of freedom depending on the length of the time series. Only correlation coefficients exceeding 95% (r $\geq$ 0.34, n=34) are shown; bold-italic values exceed 99% (r $\geq$ 0.42, n=34); bold values exceed 99.9% (r $\geq$ 0.54, n=34). SAMA and IPO have been adjusted for a lower degree of freedom (df=28) as the reconstructions end in 2007. Nss denotes 'not statistically significant'. Correlation between RICE  $\delta$ D and RICE Acc is r=(0.40) **0.49**, p<(0.05) <0.01; RICE  $\delta$ Do and RICE Acco is r=(0.45) **0.62**, p<(0.01) 0.01.

| R                       | ERAi SAT                | ERAi Precip             | SIE                       | SAMA               | SOI                   | Niño 4                | Niño 3.4              | IPO                   | NB2014                |
|-------------------------|-------------------------|-------------------------|---------------------------|--------------------|-----------------------|-----------------------|-----------------------|-----------------------|-----------------------|
| RICE δD/δD o | 0.45 /0.75       | 0.13/ 0.49       | -0.37/-0.53               | nss/-0.40          | nss/nss               | nss/nss               | nss/nss               | nss/nss               | nss/nss               |
| RICE δD/δD₀             | 0.42 /0.66       | 0.36/ 0.43       | - 0.49/ -0.58      | nss/-0.40          | nss/nss               | nss/nss               | nss/nss               | nss/nss               | nss/nss               |
| RICE Acc/Acco           | 0.60/0.34               | 0.67/0.34               | -0.56 <mark>/-0.42</mark> | - 0.46 /nss | nss/ <mark>nss</mark> | nss/ <mark>nss</mark> | nss/nss               | nss/ <mark>nss</mark> | nss/ <mark>nss</mark> |
| RICE Acc/Acco           | 0.60/ <mark>0.39</mark> | 0.67/ <mark>0.42</mark> | -0.56/- <mark>0.44</mark> | - 0.46 /nss | nss/ <mark>nss</mark> |
| ERAi SAT                | х                       | 0.66                    | -0.38                     | -0.49              | nss                   | nss                   | nss                   | nss                   | nss                   |
| ERAi Precip             | 0.66                    | х                       | -0.67                     | -0.42              | -0.49                 | 0.37                  | 0.39                  | 0.44                  | nss                   |
| SIE                     | -0.38                   | -0.67                   | х                         | 0.45               | 0.55                  | -0.48                 | -0.48                 | -0.58                 | nss                   |

Below the original and revised versions of Figures 2 and 4 are shown. In Figure 2, panels (b), (c), (d), (e), and (f) are revised. In Figure 4, panel (d) are revised.

**FIGURE 2**

Figure 1-Original: Spatial correlation fields exceeding  $\geq$  95% significance between a) ERAi annual SAT at the RICE site with ERAi annual SAT in the Antarctic / Southern Ocean region and b) ERAi annual SAT and annually averaged RICE  $\delta$ D data, c) as for b but with optimised RICE  $\delta$ D data alignment within the dating uncertainty. The correlation has been performed using ClimateReanalyzer.Org, University of Maine, USA, d) time series of ERAi SAT and RICE  $\delta$ D data, e) time series of ERAi SAT and optimised RICE  $\delta$ D data alignment, and f) scatter plot between RICE  $\delta$ D0 and ERAi SAT. The coloured dots indicate the locations of the drill sites – RICE (blue), Siple Dome (green), WDC (red), and TALDICE (pink)

---

## Author Response (AR1)

Below we outline the changes to the manuscript. First the changes in response to comments by reviewer 1 and 2 and then additional changes conducted based on our authors note. Changes are outlined below blue. References to pages and lines refer to the revised manuscript with tracked changes.

**Anonymous Referee #1**

Overall: This paper provides an analysis of a new dataset stemming from the Roosevelt Island Climate Evolution ice core. The authors identify how well this ice core compares with ERA-Int data, approximate the regional temperature and precipitation variability, and how well it compares with other proxy data in West Antarctica and portions of the western Ross Ice Shelf. The authors note an finding of a 'Ross Sea dipole' where there are periods of opposite relationships between the eastern and the western Ross Ice Shelf.

I think the paper holds promise, and certainly this new dataset needs to be presented and discussed widely. However, I find concern in interpreting the Ross Sea dipole to the SAM.

The authors base the SAM connection with the Ross Sea dipole from one paper, Marshall and Thompson (2016), presumably Fig. 2b in this paper. This figure from Marshall and Thompson (2016) indeed shows opposite patterns of heat flux across the eastern and western Ross Ice shelf associated with the SAM. However, I'm not sure that this can easily be implied as consistent with the results from the anti-phase relationships observed in the proxy data presented in the paper, for a few reasons:

a) The relationships in the Marshall and Thompson (2016) paper were based on daily data, and the authors note that the heat flux relationships with the SAM are much weaker when integrated over time periods more than a week. It is therefore really hard to know if they still exist on annual mean data (let alone data that are smoothed with 200-year moving averages!). This dipole pattern with the SAM and the heat flux is also found in reanalysis data since 1979, which certainly can't tell us much about its persistence on timescales back more than 100 years.

b) Even if there were a dipole pattern associated with the SAM that persisted, there clearly isn't a dipole pattern with temperature and the SAM (Fig. 3b of Marshall and Thompson (2016) and many other works, including Marshall (2007), Thompson and Solomon (2002) etc.). In terms of temperature, the SAM exerts the same-sign relationship across the entire Ross Ice shelf, and West Antarctica and East Antarctica. For precipitation / accumulation, there may be more of a dipole like structure (this is nearly impossible to verify with observations or reanalyses), but so many local factors influence precipitation / accumulation that it is hard to say how robust any Ross Sea dipole pattern is.

c) Climate connections with the ASL, whether from ENSO or the SAM, show dipole patterns on much larger scales, with differences (in temperature, precipitation, winds, etc) occurring between the Antarctic Peninsula / eastern West Antarctica and the Ross Sea (Ice Shelf) / western West Antarctica. I don't know fully how a dipole pattern across just the Ross Ice Shelf, from annual mean data or longer, could be related to these much larger-scale climate patterns, at least based on observations and contemporary reanalyses.

I therefore found the climate connections and their interpretation with the RICE data to be far too simplistic and an incorrect interpretation of one figure from Marshall and Thompson (2016). The authors need to revise this portion of the paper and better justify / support the pattern in relationship to the SAM, or simply not make claims that it is consistent with the SAM.

We have revised the manuscript to remove statements suggesting causality and instead only note the co-variance between the  $SAM_A$  record and the reconstructed spatial temperature pattern. This includes:

Removal of the following lines:

Page 2, ln 36-40 Page 10, ln 34-37 Page 13, ln 31-32 Page 14, ln 4-6 Page 14, ln 22-25 Page 15, ln 2-5 Page 15, ln 18-19 Page 15, ln 27-30 Page 16, ln 8-9 Page 16, ln 14-18

Page 4, ln 2-3: To clarify that the SAM / PSA induced dipole pattern is only seen in surface winds and heat flux but not in SAT, we added the following sentence "*No such pattern is observed for reginal SAT (Marshall and Thompson, 2016), which might be masked by the influence of reginal sea ice variability on local temperatures.*" That sentence is further explained by existing text that follows.

In addition, we added the  $SAM_A$  and El Niño 3.4 records (smoothed with a 200 year moving average) to Figure 8 to show the co-variance of the records but we refrain from suggesting causality.

Minor comments:

Abstract, line 31: change 'Annual' to 'Annular'

Page 2, ln 39: We have removed that sentence in response to the points raised by the reviewer above.

Line pg 3: 26: gradient of what, exactly? Just pressure / height, or other fields?

Page 3, ln 37: We have corrected the expression to "... reduced poleward pressure gradient ..."

Lines pg 3. 30-33: you should specify this is increase in total Antarctic SIE, as there are regional differences.

Pg 4, line 4: We have changed the sentence to "... at least partially to an increase in total Antarctic sea ice, while ..."

Figure 2: the color scale for the correlations is odd. It makes it challenging to see what the magnitude of the correlations are in the top panel. Even if they are significantly different from zero, a small correlation explains very little of the interannual variability and therefore may not be an ideal representation of temperature variability at other regions in West Antarctica or off the Ross Ice Shelf.

The spatial correlation pattern in panel (a), (b), and (c) are shown to highlight and compare the spatial representativeness of records derived from the RICE site (extracted ERAi data) and actual RICE data ( $\delta D$ , snow accumulation). Only correlations significant at >95% are included. While weak correlations can be useful, correlations at r> +0.4 and r< -0.4 are especially distinct with the chosen colour scale changing from red to yellow and from turquoise to blue, respectively. We feel that this colour scheme therefore provides an accessible and clear representation of both the pattern and the strength of the correlation. No changes were made.

Fig 2e, discussion of temp. trends on pg. 6 lines 37-40 and pg. 7 lines 1-7: It is fair to say ERA-Int may not capture the correct trend at the RICE site, but why not compare with observations directly at McMurdo / Scott Base or any of the longer Wisconsin AWS records (Gill , Ferrell, etc.) on the Ross Ice Shelf? These are strongly correlated with the RICE site based on Fig. 2a. I think comparing with NB2014 is helpful, but I think it is a huge oversight to not do any comparisons with direct observations (you could even use the Byrd temperature record here).

We have included Figure S2 and Table S1 in the supplementary information of the manuscript. For clarity, the comparison in Figure S2 is shown for the raw data covering 1957-2012 and for standardised data for 1957-2012 and 1979-2012. The comparison highlights that while all data sets agree on the occurrence of particularly cold (i.e. 2004, 2010) or warm (i.e. 1980) years, there is large spatial, interannual variability with respect to magnitude of change and trends across the data sets. Table S1 shows that for the satellite period (1979-2012), only the correlation between RICE  $\delta$ D and ERAi is statistically significant. If the 1957-2012 time period is considered for the NB2014 reanalysis data, then the correlation also becomes significant, but only at p<0.1, which is a level not considered in our original manuscript.

Page 7, 38-42: We have added a paragraph summarising the available AWS and station data and respective correlations in the manuscript: "We also compare the  $\delta$ Do with records from AWS (Ferrell, Gill, and Margaret AWS) and stations (Byrd, McMurdo, Scott Base, Siple Stations) in the region (Figure S2 and Table S1, supplementary information). The comparison is hampered by the shortness of the records and gaps in the observations. Only years were used for which monthly values were reported for each month of the year. No statistically significant correlation was identified between  $\delta$ Do and available data."

Page 8, In 5-7: Furthermore, we have added the sentences: "If the full time series available for the NB2014 data is considered (1957-2012), the NB2014-RICE  $\delta D_o$  correlation becomes weakly statistically significant with r=0.23 (p=0.09, Table S1)."

Fig 3, discussion page 7 lines 30-31: ERA-Int could also be different in that it uses a different snow density and/or conversion from precipitation to water equivalent. (something with the microphysics in ERA-Int model).

Page 8, In 32--35: We have added the sentence: "...and the actual drill site location, as well as differences in assumed snow densities, or different methodologies in the conversion from precipitation to water equivalent units."

Figure 4: Also not particularly happy about the color scale here for the correlations.

We make the same argument as for Figure 2 above. No changes have been made.

Page 7, lines 37-38: It would be more instructive to say that the negative correlation includes regions of the South Pacific, Antarctic Peninsula, and eastern West Antarctica, rather than the 'ASL region' since the ASL varies its location from month to month, and the correlation is not significant across the entire region that the ASL may reside.

Page 9, In 5: We have changed the wording accordingly: "A negative correlation is found in the regions of the South Pacific, Antarctic Peninsula and eastern West Antarctica"

Page 9, lines 3-5: The Nino 3.4 and Nino 4 are close (and overlap partially), and are therefore strongly temporally correlated. However PSA1 and PSA2, by design through EOF, are uncorrelated in time and space. I don't think using Nino 4 for PSA2 is a good idea because of this.

Page 10, 14-18: We have removed the association of PSA1 and PSA2 with Niño 3.4 and Niño 4. "In addition, the Southern Oscillation Index (SOI, Trenberth and Stepaniak, 2001), Niño 3.4 Index (Rayner et al., 2003b), Niño 4 Index (Trenberth and Stepaniak, 2001), and the IPO Index (Henley et al., 2015) are used to investigate the influence of SST variability in the eastern and central tropical Pacific on annual and decadal time scales (IPO)."

Page 9, Table 2 ENSO correlations lines 10-15: In addition to differences in the phasing of ENSO and SAM, using annual means for ENSO is also compromising the correlations, since ENSO events wrap

around a calendar year (peaking in December often). They are likely stronger on seasonal means; this should be mentioned.

Page 10, ln 32-34: We added the following sentence "*Moreover, statistically significant correlations might also be obtained if seasonal averages could be used for the comparison as ENSO events usually peak during the austral summer, in particular December (Turner, 2004).*" and have added the following reference: The El Niño–Southern Oscillation and Antarctica, International Journal of Climatology, 24, 1-31, 10.1002/joc.965, 2004.

**Anonymous Referee #2**

In this study, the authors are presenting high-resolution isotopic (dD) and snow accumulation rate records of a new ice core (RICE) drilled on the Roosevelt Island in the eastern Ross Sea and covering in its upper part the last 2700 years. The authors are comparing these new records to other ice core records present in the near-by areas interpreting these records in terms of the climate variability which in these areas (Eastern and Western Ross Sea) is characterized by a climate pattern referred to as the Ross Sea dipole. The paper is noteworthy and the authors are doing a good job in calibrating the new records against the Era-interim re-analysis data (temperature and precipitation) as well as to other climate indexes as SAM, SOI, Nino3.4, IPO, as well as Sea Ice extent. However, some methodological questions are arising in this part (see C1 below). The manuscript is quite well structured and the topic is appropriate for Climate of the Past. Nevertheless, the authors should consider the comments reported below before resubmitting a revised version.

One general comments refers to the fact that the manuscript (Winstrup et al., CPD) presenting the ice core dating, on which most of the interpreted data are relying on, is not published yet.

This publication is now in review and can be accessed here: https://www.clim-past-discuss.net/cp-2017-101/ including comments from two reviewers. We have updated the reference.

There are also other papers (one is Emanuelsson et al.), to which the authors are referring that are still at the submission7review stage, please check and update.

Three publications referred to in our manuscript are still in review: Emanuelsson et al. (in review), Keller et al. (in review), and Pyne et al. (in review). The references have been updated.

General as well as detailed comments are reported below.

Page 3, line 3: may you check this sentence? Over the observational period (satellite era) the sea ice should be increasing in the Ross Sea sector and decreasing in the Amundsen–Bellingshausen sector.

Page 3, ln 13-14: We have corrected the sentence to state "...changes in sea ice (wind driven, regional decreases and increases in the Amundsen and Ross Seas, respectively."

Page 3, line 11: here the authors are saying that they will compare these new records to other ones existing in the region but it is not clear why they do not consider the Taylor Dome ice core record.

We have included the Taylor Dome stable isotope record in the discussion in section "5.1 - RegionalTemperature Variability" and Figure 6. We did not include the Taylor Dome data in sections 6, 6.1, 6.2, 6.3 or in Figure 7, 8 and 9 because of its age scale uncertainty.

Page 11, ln 3-4: We have introduced the Taylor Dome data "... coastal East Antarctica (Talos Dome, TALDICE – purple, Taylor Dome – orange) and West Antarctica ..."

Page 11, ln 8-9: "... exhibit a long-term cooling trend for West Antarctica, while TALDICE (Stenni et al. 2012) and Taylor Dome (Steig et al. 1998, Steig et al. 2000) recorded stable isotopic conditions ..."

Page 12, In 5-8: "The TALDICE and Taylor Dome water stable-isotope temperature do not exhibit a long term trend over the past 2.7ka. Yet colder water stable –isotope temperature anomalies have been associated with the LIA period at both sites (Stenni et al. 2012) ..."

Page 15, ln 41: "... appeared decoupled (TALDICE/Taylor Dome and Talos Dome), no trend in isotope or precipitation, respectively)."

Page 6, lines 27-32. Here the authors are optimizing the dD/T relation to the age scale. I am wondering why the authors did not consider optimizing the snow accumulation rate to the ERA-I precipitation rather than the dD/T relation. In fact, it is known and also the authors are clearly showing this, that the isotopic composition rely not only on the temperature but also on other circulation-related factors. How, this choice is affecting the climate interpretation? The authors should answer to this comment.

Page 9, ln 11-14: We have added the following statements: "This result suggests that the optimised age solution is not superior to the RICE17 age scale and we note that the sensitivity of the correlation to those minor adjustments is founded in the brevity of the common time period. However, there is no significant difference between the overall pattern and relationships of the two age scale solutions."

Page 6, line 31: Figure 2b should be 2c.

Page 7, ln 26: Done

Page 6, line 35: in the Masson-Delmotte paper only spatial d/T slope are considered.

Page 7, ln 29-31: We have broadened the cited  $\delta/T$  slopes to include the range reported by Schneider et al. 2005 (interannual  $\delta/T$  slopes): "From the comparison between RICE  $\delta$ Do and ERAi SAT records, we obtain a temporal slope of 3.37 ‰ °C-1 (Figure 2f), which falls within the limit of previously reported values from Antarctica of ~2.90 – 3.43 ‰ °C-1 for temporal (interannual) slopes (Schneider et al., 2005) and  $\pm 0.51$  ‰ °C-1 to ~6.80  $\pm 0.57$  ‰ °C-1 for spatial slopes (Masson-Delmotte et al., 2008). We use this relationship to calculate temperature variations for the RICE  $\delta$ D record."

Page 7, line 5: Is the lack of correlation valid also considering only the 1979-2012?

Page 7, In 38-42: We have added the following paragraph: "We also compare the  $\delta Do$  with records from AWS (Ferrell, Gill, and Margaret AWS) and stations (Byrd, McMurdo, Scott Base, Siple Stations) in the region (Figure S2 and Table S1, supplementary information). The comparison is hampered by the shortness of the records and gaps in the observations. Only years were used for which monthly values were reported for each month of the year. No statistically significant correlation was identified between  $\delta Do$  and available data."

Page 8, In 5-8: We have added the following sentence: "If the full time period available for the NB2014 data is considered (1957-2012), the NB2014 – RICE  $\delta$ Do correlation becomes weakly statistically significant with r=0.23 (p=0.09, Table S1)."

Page 10, ln 28-34: We have added the following sentence: "In 2010, an anomalously cold year is observed. If only the time series from 1979-2009 is considered, the correlations between RICE  $\delta D$  and these considered climate drivers becomes statistically significant: SOI (r=-0.48, p=0.006), Niño 3.4 (r=0.48, p=0.007); Niño 4 (r=0.57, p=0.001), and IPO (r=0.44, p=0.014). This further highlights the vulnerability of this analysis to individual years due to the brevity of the time series further complicating a linear analysis between individual drivers and regional responses."

Page 7, line 6: RICE dD should be RICE dDo.

Page 8, ln 5: We corrected it to "δDo"

Page 7, lines 11-14: Is this strong accumulation rate gradient suggesting possible movements of the dome in the past. May you explain this?

Page 5, In 17-22: We have added the following paragraph: "A small migration of the divide has occurred in the past few centuries with the topographic divide off-set by about 500 m to the southwest. It is possible that the divide migrated as a result of an imbalance in the ice flux on either side of the divide, by either changes in the snow accumulation gradient or changes in the efflux across the grounding-line due perhaps to changes in the buttressing by the Ross Ice Shelf. However, the negligible divide position migration magnitude suggests neither snow accumulation gradient nor grounding line efflux have changed very much; this implies that the buttressing has not changed significantly either."

Page 7, line 37-38: the region which exhibits a negative correlation seems to be at lower latitudes than the ASL, at least looking at the figure . . ..

Page 9, In 5: We have changed the sentence to "A negative correlation is found in the regions of the South Pacific, Antarctic Peninsula and the eastern West Antarctica".

Page 8, lines 10-12: the strong impact of blocking events at this site would support my comment above (Page 6, lines 27-32).

As described in our response to the reviewer, ERAi precipitation and RICE snow accumulation correlate at r=0.60. Using the same optimisation methodology as tested for the RICE  $\delta D$  record only marginally improves the correlation (r=0.62). For this reason, no optimisation was carried out.

Page 8, line 13: the negative correlation seems to interest more the Amundsen Sea. Page 8, line 17: are polynyas resolved by the model data?

Page 9, In 24: We have clarified the paragraph: "Snow accumulation at RICE is negatively correlated with sea ice concentration (SIC) in the Ross Sea region and northern Amundsen Sea region (Figure 5a), which predominately represents sea ice exported from the Ross Sea. We observe that years of increased (decreased) SIC leading to reduced (increased) accumulation at RICE, confirming the sensitivity of moisture-bearing marine air mass intrusions to local ocean moisture sources and hence regional SIC. The correlation between ERAi SIC and the optimised RICE  $\delta$ Do record (Figure 5b) similarly shows a negative correlation of SIC in the Ross Sea (perhaps with the exception of the Ross and Terra Nova polynyas) and the northern Amundsen Sea suggesting more depleted (enriched) values during years of increased (reduced) SIC."

Page 8, line 21: I would move this sentence: "We focus . . .." at the beginning of the paragraph.

Page 9, ln 31-35: We have replaced the paragraph with "In Figure 5 c and d, the sea ice extent index (SIEJ) for the Ross-Amundsen Sea, developed by Jones et al. (2016), is correlated with ERAi SAT and precipitation data."

Page 8, line 20: please add C2 a URL link to these data (SIEj).

Page 16, ln 22-43 and Page 17, ln 1-9: We have added the sources (URLs) for all RICE data, meteorological observations and climate indices used in this manuscripts.

Page 9, lines 1-7: please add data citation or URL in Data Availability section for all the climate indexes used.

Please see comment above.

Page 9, line 9: why using the SAMa index instead of SAM for this period? Are these two indexes the same over this period?

Page 10, ln 14-15:We added the following sentence: "We use the SAMA index developed by Abram et al. (2014) to test the fidelity of the SAM relationship with the climatic conditions in the Ross Sea over the past millennium (Table 2). The SAMA is highly correlated (r=0.75) with the SAM record developed by Marshall (2003) for their common time period (1957-2009)."

Page 9, line 13: . . ... (but not with Rice snow accumulation). . . See again my comment above (Page 6, lines 27-32).

Page 9, ln 11-14: We have added the following statements: "This result suggests that the optimised age solution is not superior to the RICE17 age scale and we note that the sensitivity of the correlation to those minor adjustments is founded in the brevity of the common time period. However, there is no significant difference between the overall pattern and relationships of the two age scale solutions."

Page 9, lines 22-24: again why not using Taylor Dome? On the other side, regarding the use of TALDICE data: I would suggest to use Talos Dome (89 m core) isotopic and snow accumulation rate data (Stenni et al., 2002) rather than TALDICE for this recent period. The TALDICE data are low resolution data and the snow accumulation rate, here considered, comes from the dating model and as such implicitly connected to the isotopic record from which it is derived. On the contrary, the isotopic and snow accumulation rate records from the Talos Dome core, although limited to the past 800 years, are high-resolution data and the dating has been performed by nssSO4 annual data constrained by the volcanic chronology. Moreover, the TALDICE data, here reported are on the Buiron et al. (2011) age scale which has been replaced by the AICC-2012 chronology. Between the two there are in some cases differences up to 150 years than for the purposes of this paper could be important. So, the authors could consider using both isotopic data sets (Talos Dome and TALDICE but the latter on the AICC-2012 age scale) and the snow accumulation rate from only Talos Dome. An alternative to the AICC2012 age scale for TALDICE is the Severi et al. (2012, CP) chronology, which uses the volcanic synchronisation between the TALDICE and the EPICA Dome C ice cores and in practice identical to AICC2012 age scale.

We have replaced the TALDICE (a) snow accumulation and (b)  $\delta^{18}$ O records on the Buiron et al. 2011 age scale with (a) the Talos Dome (TD96) record for snow accumulation on the Severi et al. 2012 age scale and (b) TALDICE  $\delta^{18}$ O on the Severi et al. 2012 age scale for the temperature reconstruction. We have updated the relevant figures (Figure 6, 8, and 9).

Page 25, Table 1: We included in the table the age scale used for all ice core records used in this manuscript.

Page 10, line 14: the onset of a decline in WDC isotopes at 579 CE is not clear from the figure . . .

Page 11, ln 40 to Page 12, ln 1: We have revised the sentence to "We note that within 200 years of the onset of the isotopic warming at RICE (at 580 CE  $\pm$  27 years), the WDC borehole temperature and isotope data start to record a temperature decline, in line with the observed anti-phased relationship of WDC with RICE and Siple Dome."

Page 10, lines 25-26: the sentence "From the 17th century . . . . . " it is not very clear, at least looking at the figure, and the word in phase seems to be not the correct word to use. . ..

Page 12, ln 11-17: We have revised this sentence to "Overall this comparison shows that isotope temperature trends in the eastern Ross Sea (isotopic warming at RICE and Siple Dome) and West Antarctica (WDC cooling) were anti-phased for over 2 ka (660 BCE to ~1500 CE), while the western Ross Sea (TALDICE) remained stable, forming a distinct Ross Sea Dipole pattern. From the 17th Century onwards this relationship changes. While WDC water stable isotope temperatures continue to cool, from the 17th Century, the WDC borehole temperature records a warming. At the same time,

*RICE and Siple Dome experience warmer isotope temperatures while TALDICE recovers from its coldest recorded isotope temperature during the study period.*"

**Additional Revisions to the Manuscript**

As outlined in our authors note, we erroneously used in the comparison between the reanalysis and RICE data which stemmed from the annually resampled RICE  $\delta D$  record instead of the annually averaged RICE  $\delta D$  record. However, correctly using the annually averaged RICE  $\delta D$  record does not change our interpretation or conclusions.

We have revised all relevant section in the manuscript to now include the correct annually averaged RICE  $\delta D$  data and revised all associated correlation values. The comparison with the correct data shows that in most cases the correlations either did not change or improved, with exception of the correlation between RICE  $\delta D/\delta Do$  and SAT/Precipitation where correlations weakened. Original values that required re-calculation are shown in **'red'** while the new, revised values are shown in **'blue'**

**Table 1:** Overview of correlation coefficients for annual means of the common time period 1979-2012 between climate parameters, proxies and indices: the original RICE ( $\delta$ D) and optimised ( $\delta$ Do) data (this paper), the original RICE snow accumulation data (RICE Acc, Winstrup et al., in review) and data adjusted to the revised age scale of  $\delta$ Do – Acco, ERAi Surface Temperature (ERAi SAT) and Precipitation (ERAi Precip), Dee at el., 2011), Ross/Amundsen Sea Sea Ice Extent (SIEJ, Jones et al., 2016), Southern Annular Mode Index (SAMA, Abram et al., 2014), Southern Oscillation Index (SOI, Trenberth and Stepaniak, 2001), Niño 4 Index (Trenberth and Stepaniak, 2001) and Niño 3.4 (Emile-Geay et al., 2013), Interdecadal Pacific Oscillation Index (IPO, Henley et al., 2015), and the near-surface Antarctic temperature reconstruction (NB2014, Nicolas and Bromwich, 2014). Significance values are adjusted for degree of freedom depending on the length of the time series. Only correlation coefficients exceeding 95% (r $\geq$ 0.34, n=34) are shown; bold-italic values exceed 99% (r $\geq$ 0.42, n=34); bold values exceed 99.9% (r $\geq$ 0.54, n=34). SAMA and IPO have been adjusted for a lower degree of freedom (df=28) as the reconstructions end in 2007. Nss denotes 'not statistically significant'. Correlation between RICE  $\delta$ D and RICE Acc is r=(0.40) 0.49, p<(0.05) <0.01; RICE  $\delta$ Do and RICE Acco is r=(0.45) 0.62, p<(0.01) 0.01.

| R                       | ERAi SAT          | ERAi Precip             | SIE                       | SAMA               | SOI                   | Niño 4                | Niño 3.4              | IPO                   | NB2014                |
|-------------------------|-------------------|-------------------------|---------------------------|--------------------|-----------------------|-----------------------|-----------------------|-----------------------|-----------------------|
| RICE δD/δD o | 0.45/0.75         | 0.13/ 0.49       | -0.37/-0.53               | nss/-0.40          | nss/nss               | nss/nss               | nss/nss               | nss/nss               | nss/nss               |
| RICE δD/δD o | 0.42/0. 66 | 0.36/ 0.43       | - 0.49/- 0.58      | nss/-0.40          | nss/nss               | nss/nss               | nss/nss               | nss/nss               | nss/nss               |
| RICE Acc/Acco           | 0.60/0.34         | 0.67/0.34               | -0.56 <mark>/-0.42</mark> | - 0.46/nss  | nss/ <mark>nss</mark> |
| RICE Acc/Acco           | 0.60/ 0.39 | 0.67 <mark>/0.42</mark> | -0.56 <mark>/-0.44</mark> | - 0.46 /nss | nss/ <mark>nss</mark> |
| ERAi SAT                | х                 | 0.66                    | -0.38                     | -0.49              | nss                   | nss                   | nss                   | nss                   | nss                   |
| ERAi Precip             | 0.66              | х                       | -0.67                     | -0.42              | -0.49                 | 0.37                  | 0.39                  | 0.44                  | nss                   |
| SIE                     | -0.38             | -0.67                   | х                         | 0.45               | 0.55                  | -0.48                 | -0.48                 | -0.58                 | nss                   |

Page 6, In 36-40 to page 7, In 1-12: We have added a brief discussion on the average annual temperature derived from ERAi, AWS and borehole temperature measurements: "The borehole temperature *measured in 2012 in two 11 m and 12 m deep drill holes suggest an average annual temperature of - 23.5 °C. This stands in stark contrast to the average annual temperature derived from ERAi data of - 27.4 °C at the RICE site. Furthermore, the RICE AWS recorded also an average annual temperature of also -23.5 °C. In contrast, the nearby Margaret AWS (located at 67m above sea level, just 96 km southwest of the RICE AWS, data obtained from Antarctic Meteorological Research Center and Automatic Weather Station Project; https://amrc.ssec.wisc.edu) records an average annual temperature of -26.6 °C. While recorded summer temperatures at RICE and Margaret AWS agree well, during winter the Margaret AWS records up to 10-15 °C colder 3-hourly temperatures than the RICE AWS. It is possible that rime build-up at the RICE AWS during winter (Supplementary Information, Figure S1) might have provided insulation that allowed for residual heat from the sensors to warm the temperature cavity leading to erroneously warm readings. Alternatively, it is possible that the Margaret AWS site is*

influenced by stronger temperature inversions leading to exceptionally cold temperatures of -60 °C, while the topography of Roosevelt Island might be less conducive to such conditions. A comparison between high resolution borehole temperature measurements conducted at RICE from November 2013 to November 2014 and AWS data, including snow temperature measurements, will provide important insights into this temperature off-set. Until this analysis is concluded, we argue that ERAi data, which agree well with the Margaret AWS observations, provide the most reliable temperature time series to calibrate the stable isotope – temperature relationship."

Furthermore we have revised:

Figure 1: We have added the location of Taylor Dome

Figure 2: We have revised the figures to now show the annually averaged  $\delta D$  and  $\delta Do$  data and show the location of Taylor Dome

Figure 3: No change

Figure 4d: We have revised this panel to show the time series using the age scale derived from the revised  $\delta Do$  age scale and show the location of Taylor Dome

Figure 5: We have added the location of Taylor Dome

Figure 6: We have added the Taylor Dome stable isotope record and added the Taylor Dome location. We have revised the TALDICE data to show the TALDICE stable isotope data for the temperature reconstruction and the Talos Dome snow accumulation data. Both are plotted on the Severi et al.2012 age scale. We have revised the change points for the RICE  $\delta D$  data which shifted from 578 CE to 580 CE and from 1491 CE to 1477 CE with the use of the correct average annual averaged RICE  $\delta D$  record.

Figure 7: We have revised the temperature calculation and revised the change points.

Figure 8: We have revised the RICE  $\delta D$  record to the annually averaged RICE  $\delta D$ , used the TALDICE stable isotope data on the Severi et al. 2012 age scale and added the normalised, smoothed SAMA and Niño 3.4 reconstructions.

Figure 9: We have revised the RICE  $\delta D$  record to the annually averaged RICE  $\delta D$  and used the TALDCIE stable isotope data on the Severi et al.2012 age scale.

In addition we added the following Figures and Tables in the Supplementary Information:

Table S1: Pearson correlation coefficient (r) and significance values (p) for correlations between RICE  $\delta D$  record and relevant observational records from automatic weather stations and reanalysis data. Data for the reconstructed Byrd Station meteorological data (Bromwich et al., 2013) are accessed via the Byrd Polar Research Centre, Polar Meteorological Group, Ohio State University (http://www.polarmet.osu.edu/datasets/Byrd\_recon/). Weather station data for Ferrell, Gill, and Margaret AWS are accessed via Antarctic Meteorological Research Center and Automatic Weather Station Project (https://amrc.ssec.wisc.edu). Data for McMurdo Station and Scott Base are accessed via the MET-READER (https://legacy.bas.ac.uk/met/READER/data.html). The number of years of

observations represents the total number of years which contain monthly averages for each month of a calendar year. Only years with 12 monthly values are included in the correlation.

Figure S1: RICE AWS covered with rime after the winter season. The photo was taken in November 2012.

Figure S2: Comparison of temperature data from Antarctic Stations, remote Antarctic Weather Stations (AWS), and reanalysis products with  $\delta D$  RICE data. Origin of the data is referenced in Table R1-1. Panel (a) shows the actual data, panels (b) and (c) show the comparison of the standardised records for the time periods 1957-2012 and 1975-2012, respectively.

Finally, some very minor revisions to the text were made (tracked changes) for consistency and to streamline the manuscript.

**The Ross Sea Dipole - Temperature, Snow Accumulation and Sea Ice Variability in the Ross Sea Region, Antarctica, over the Past 2,700 Years**

**5 **RICE Community**

35

(Nancy A.N. Bertler1,2, Howard Conway3, Dorthe Dahl-Jensen4, Daniel B. Emanuelsson1,2, Mai Winstrup4, Paul T. Vallelonga4, James E. Lee5, Ed J. Brook5, Jeffrey P. Severinghaus6, Taylor J. Fudge3, Elizabeth D. Keller2, W. Troy Baisden22,7, Richard C.A. Hindmarsh78, Peter D. Neff81,2,9, Thomas Blunier4, Ross Edwards910, Paul A. Mayewski1011, Sepp Kipfstuhl1+12, Christo Buizert5, Silvia Canessa2, Ruzica Dadic1, Helle A. Kjær4, Andrei Kurbatov1011, Dongqi Zhang12,13,14, Edwin D. Waddington3, Giovanni Baccolo1415, Thomas Beers1011, Hannah J. Brightley1,2, Lionel Carter1, David Clemens-Sewall1516, Viorela G. Ciobanu4, Barbara Delmonte1415, Lukas Eling1,2, Aja A.–Ellis1617, Shruthi Ganesh1418, Nicholas R. Golledge1,2, Skylar Haines4011, Michael Handley4011, Robert L. Hawley1516, Chad M. Hogan1819, Katelyn M. Johnson1,2, Elena Korotkikh4011, Daniel P. Lowry1, Darcy Mandeno1, Robert I. Markay1, James A. Menking5, Timothy R. Naish1, Caroline Noerling1412, Agathe Ollive4920, Anaïs Orsi2021, Bernadette C. Proemse1819, Alexander R. Pyne1, Rebecca L. Pyne2, James Renwick1, Reed P. Scherer2422, Stefanie Semper2223, M. Simonsen4, Sharon B. Sneed4011, Eric J., Steig3, Andrea Tuohy2312,2,4, Abhijith Ulayottil Venugopal1,2, Fernando Valero-Delgado4412, Janani Venkatesh4718, Feitang Wang2425, Shimeng Wang4314, Dominic A. Winski4516, Victoria V. H-olly L. Winton2526, Arran Whiteford2627, Cunde U. 1920

20 Xiao2728, Jiao Yang1314, Xin Zhang2829)

[revised manuscript text omitted]

1,000 years  $\leq \pm 19$  years and for the past 2,000 years  $\leq \pm 38$  years, reaching a maximum uncertainty of  $\pm 45$  years at 344 m depth (2.7 ka). The RICE17 timescale is in good agreement with the WD2014 annual-layer counted timescale from the WAIS Divide ice core dating to 200 Common EraCE (CE, 280 m depth). For the deeper parts of the core, there is likely a small bias (2-3,%) towards undercounting the annual layers, resulting in a small age offset compared to the WD2014 timescale-(Winstrup et al., 2017).

5

**3.2 Snow accumulation reconstruction**

Ice core annual layer thicknesses provide a record of past snow accumulation once the amount of vertical strain has been accounted for. At Roosevelt Island, repeat pRES measurements were performed across the divide, providing a direct measurement of the vertical velocity profile (Kingslake et al., 2014). This has a key advantage over most previous ice-core 10 inferences of accumulation rate because vertical strain thinning through the ice sheet is measured directly, rather than needing to use an approximation for ice-flow near ice divides (e.g. Dansgaard and Johnsen, 1969; Lliboutry, 1979). Uncertainty in the accumulation-rate reconstruction increases from zero at the surface (no strain thinning) to a maximum of  $\pm 8.10$  % at  $\frac{170.78}{10.000}$ m true depth (1712 CE). Below  $\frac{170-78}{7}$  m, the uncertainty remains constant at  $\pm \frac{8}{10}$ %. 
[revised manuscript text omitted]
, 2016). We do not interpret the Taylor Dome record for this time period because its age scale uncertainties in this part of the record (0-3ka BP is based on a flow model which assumes constant snow accumulation and lacks independent age benchmarks, Steig et al. (1998), Steig et al. (2000)(Steig et al.,
- 35 1998;Steig et al., 2000)) prohibits a precise alignment of events(Steig et al., 1998;Steig et al., 2000). 
[revised manuscript text omitted]

|                    |                |                 |                  |                       | al. 2012  |                   | 2011, Severi et al. 2012             |  |  |
| Talos Dome         | S 72.80 | E 159.06 | 2,316     | 89             | Severi et        | 1995       | Stenni et al. 2002            |  |  |
| (TD96)      |                |                 |                  |                       | al. 2012  |                   |                                      |  |  |
| Taylor Dome | S 77.70        | E 159.07        | 2,375            | 554                   | st9810           | 1993-1994         | Steig et al. 1998, Steig et al. 2002 |  |  |

Table 1: Overview of ice core records used in this manuscript. Locations are present in Figure 1.

5

Table 2: Overview of correlation coefficients for annual means of the common time period 1979-2012 between climate parameters, proxies and indices: the original RICE (δD) and optimised (δD₀) data (this paper), the original RICE snow accumulation data (RICE Acc, Winstrup et al., in preparation2017) and data adjusted to the revised age scale of δD₀ – Acc₀, ERAi Surface Temperature (ERAi SAT) and Precipitation (ERAi Precip), Dee at el., 2011), Ross/Amundsen Sea Sea Ice Extent (SIEJ, Jones et al., 2016), Southern Annular Mode Index (SAM₄, Abram et al., 2014), Southern Oscillation Index (SOI, Trenberth and Stepaniak, 2001), Niño 4 Index (Trenberth and Stepaniak, 2001) and Niño 3.4 (Emile-Geay et al., 2013), Inter-decadal Pacific Oscillation Index (IPO, Henley et al., 2015), and the near-surface Antarctic temperature reconstruction (NB2014, Nicolas and Bromwich, 2014). Significance values are adjusted for degree of freedom depending on the length of the time series. Only correlation coefficients exceeding 95% (r≥0.34, n=34) are shown; bold-italic values exceed 99% (r≥0.42, n=34); bold values exceed 99.9% (r≥0.54, n=34). SAM₄ and IPO have been adjusted for a lower degree of freedom (df=28) as the reconstructions end in 2007. Nss denotes 'not statistically significant'. Correlation between RICE δD and RICE Acc is r=0.400.49, p<0.050.01; RICE δD₀ and RICE Acc₀ is r=0.4562, p<0.01.</li>

| R                       | ERAi SAT                                   | ERAi Precip                                | SIE J   | SAMA              | SOI     | Niño 4  | Niño 3.4 | IPO     | NB2014  |
|-------------------------|--------------------------------------------|--------------------------------------------|--------------------|-------------------|---------|---------|----------|---------|---------|
| RICE δD/δD o | <del>0.450.42/0.750.66</del> | <del>0.130.36/0.490.43</del> | - <del>0.37_</del> | nss/-0.40         | nss/nss | nss/nss | nss/nss  | nss/nss | nss/nss |
|                         |                                            |                                            | 0.49 /-     |                   |         |         |          |         |         |
|                         |                                            |                                            | 0.5 38      |                   |         |         |          |         |         |
| RICE Acc/Acco           | 0.60 /0. <del>3</del> 4 39   | 0.67 /0. 3442                | -0.56/-            | -0.46 /nss | nss/nss | nss/nss | nss/nss  | nss/nss | nss/nss |
|                         |                                            |                                            | 0.4 244     |                   |         |         |          |         |         |
| ERAi SAT                | х                                          | 0.66                                       | -0.38              | -0.49             | nss     | nss     | nss      | Nss     | nss     |
| ERAi Precip             | 0.66                                       | Х                                          | -0.67              | -0.42             | -0.49   | 0.37    | 0.39     | 0.44    | nss     |
| SIE J        | -0.38                                      | -0.67                                      | Х                  | 0.45              | 0.55    | -0.48   | -0.48    | -0.58   | nss     |

**Figures**

---

## Author Response (AR2)

Below we outline our response in blue to the comments of Reviewer #1 and #2

**Response to Reviewer #1**

The paper has improved considerably. I appreciate the authors' detail in their response and their caution to interpreting the dipole pattern seen in the ice core data to the SAM. I think the addition of the observations add a lot to the paper. I'm happy to have the paper accepted for publication, with only small technical / grammatical fixes as indicated below.

Page 3, Line 27: Remove the word 'along'

Done

Page 13, Lines 6-7: It isn't clear how a weakening of the ASL is related to fewer blocking events, which are typically done by higher pressure. A weakening of the ASL would seem to suggest more regional blocking in the ASL area. Perhaps you're talking about some other area of blocking connected to the ASL (like SE of NZ or in the S. Atlantic) that tends to be opposite-sign of the ASL due to the PSA-type wavetrains? Please rephrase this sentence to improve clarity.

The ridging relevant for RICE precipitation occurs west of the region of the ASL because of the location of the ASL. When the ASL weakens, the region of ridging either shifts or occurs less frequently. We have clarified the sentence: "The reduction in snow accumulation might be linked to a negative SAM-induced weakening of the ASL, perhaps leading to the development of fewer blocking events in the eastern Ross Sea."

Page 14, line 21: insert word 'time' before dependent

Done

Page 14, line 35: change 'experiencing' to 'all experience'

Done

**Response to Reviewer #2**

The revised version of the manuscript is much improved and I would only suggest some minor/technical modifications as summarized below:

One comment regard the Taylor Dome data: I agree that using the st9810 age scale makes no sense but why not using the updated age scale as reconstructed by by Michael Sigl (Sigl et al., 2017 Nature Clim Change, see their figure 6 in the SOM) using the volcanic matches with the WDC06A-7 chronology

of the WAIS ice core? These data are available in the PAGES2k Consortium 2017 paper: A global multiproxy database for temperature reconstructions of the Common Era. Scientific Data, 4, 170088. doi: 10.1038/sdata.2017.88. The data are available at: https://www.ncdc.noaa.gov/paleo-search/study/21171. In case this data set is used in the figure 6, please also update the text lines 34-36 at page 12 of the revised manuscript.

We thank the reviewer and have changed the age scale of the Taylor Dome data to the WDC06A-7 age scale as developed by Sigl et al.2014 and published by PAGES2k Consortium 2017. We updated Figure 6 and Table 1 to reflect these changes.

2)      For Talos Dome firn core: the dating is from Stenni et al. 2002 and not Severi et al., 2012, which is dating that can be used for the TALDICE deep ice core for the recent period. Please, update the Table 1.

Done

Other minor/technical issues:

Page 6, line10: the citation Keller et al., is lacking in the References list.

Done

Page 7, line 17: a value is lacking before +/- 0.51 ….

We removed this value which had been replaced with a new value in the revision.

Page 12, lines 34-36: update this sentence if using the Taylor Dome data on the Sigl age scale (see above).

We replaced the following sentence: ”We do not interpret the Taylor Dome record for this time period because its age scale uncertainties in this part of the record (0-3ka BP is based on a flow model which assumes constant snow accumulation and lacks independent age benchmarks, Steig et al. (1998), Steig et al. (2000)) prohibits a precise alignment of events. “ with: “The Taylor Dome record also shows prolonged cold isotope temperature anomalies during this time period.”

Page 15: in data availability update the data citation for the ice core data used in the manuscript and figures (Talos, TALDICE, Taylor Dome, Siple Dome WDC).

Done

Figure 6 caption: add Taylor Dome in part a).

Done